# CERTIFIED VS. EMPIRICAL ADVERSARIAL ROBUSTNESS VIA HYBRID CONVOLUTIONS WITH ATTENTION STOCHASTICITY

**Joy Dhar[1], Song Xia**[*2]**, Manish Kumar Pandey**[*3]**, Maryam Haghighat[4], Azadeh Alavi[5],
Ferdous Sohel[6], Wenyu Zhang**[†]**, Nayyar Zaidi[7]**

[1]Indian Institute of Technology Ropar  [2]Nanyang Technological University  [3]RoentGen Health
[4]Queensland University of Technology  [5]RMIT University  [6]Murdoch University  [7]Deakin University

## ABSTRACT

We introduce *Hybrid Convolutions with Attention Stochasticity* (`HyCAS`), an adversarial defense that narrows the long-standing gap between *provable* robustness under $\ell_2$ certificates and *empirical* robustness against strong $\ell_\infty$ attacks, while preserving strong generalization across *diverse imaging benchmarks*. `HyCAS` unifies deterministic and randomized principles by coupling 1-Lipschitz, spectrally normalized convolutions with two stochastic components—*spectral normalized random-projection filters* and a *randomized attention-noise mechanism*—to realize a *randomized defense*. Injecting smoothing randomness *inside* the architecture yields an overall $\leq 2$-Lipschitz network with formal certificates. Extensive experiments on diverse imaging benchmarks—including `CIFAR-10/100`, `ImageNet-1k`, `NIH Chest X-ray`, `HAM10000`—show that `HyCAS` surpasses prior leading certified and empirical defenses, boosting certified accuracy by up to $\approx 7.3\%$ (*on NIH Chest X-ray*) and empirical robustness by up to $\approx 3.1\%$ (on `HAM10000`), without sacrificing clean accuracy. These results show that a *randomized Lipschitz constrained architecture can simultaneously improve* both certified $\ell_2$ and empirical $\ell_\infty$ adversarial robustness, thereby supporting safer deployment of deep models in high-stakes applications. Code: https://github.com/misti1203/HyCAS

## 1 INTRODUCTION

Despite their impressive accuracy, deep learning architectures in computer vision remain vulnerable to adversarial attacks. Such vulnerabilities threaten safety-critical deployments in fraud detection (Pumsirirat & Liu, 2018), autonomous driving (Cao et al., 2021), and clinical decision support (Dhar et al., 2025), where mistakes carry high costs. In response to these adversarial vulnerabilities, early research focused on *empirical* defences, most notably adversarial training (Madry et al., 2018; Dhar et al., 2025; Ding et al., 2019; Shafahi et al., 2019; Sriramanan et al., 2021; Cheng et al., 2023; Dhar et al., 2026). However, these methods are frequently broken by intricately crafted adversarial attacks (Carlini & Wagner, 2017; Yuan et al., 2021; Hendrycks et al., 2021; Duan et al., 2021; Li et al., 2023). This limitation has fuelled interests in *certified* robustness techniques, which offer provable guarantees that a classifier's prediction cannot change within a specified perturbation radius (Raghunathan et al., 2018; Wong & Kolter, 2018; Hao et al., 2022).

Randomized Smoothing (`RS`) (Lécuyer et al., 2019; Cohen et al., 2019) certifies robustness by averaging a model's predictions over noise-perturbed inputs at inference, and therefore scales to modern deep architectures. Yet `RS` is inherently rigid: large noise budgets erode clean accuracy, whereas small budgets certify only narrow $\ell_2$ radii. Recent baselines seek to bypass this trade-off with *test-time* adaptations—both generic (Croce et al., 2022) and `RS`-specific (Alfarra et al., 2022b; Súkeník et al., 2022; Hong et al., 2022). These defences, however, are mostly heuristic-based and they quickly succumb to stronger, tailored attacks (Croce et al., 2022; Alfarra et al., 2022a; Hong et al., 2022), rekindling the familiar "cat-and-mouse" cycle of empirical defences (Athalye et al., 2018;

---

[*]Equal contribution; [†] Independent researcher.

Tramèr et al., 2020). Moreover, they are rarely benchmarked against state-of-the-art empirical attacks—such as `APGD` (Croce & Hein, 2020) or AutoAttack (Croce & Hein, 2020)—or on domain-specific distributions, such as medical-imaging datasets, thereby leaving their real-world efficacy uncertain. We move beyond pure test-time fixes and inject *fresh, independently drawn noise during **both** training and inference*. This two-phase strategy (i) preserves `RS`'s formal guarantees, (ii) alleviates the accuracy–robustness trade-off, and (iii) is validated against both certified and strong empirical attacks across *diverse imaging benchmarks* [1].

To bridge the gap between certified and empirical defenses, we introduce **Hybrid Convolutions with Attention Stochasticity (`HyCAS`)**. `HyCAS` offers provable $\ell_2$ adversarial robustness, boosts empirical adversarial resilience to strong $\ell_\infty$ attacks, and generalizes across *eight diverse vision benchmarks. It is a randomized defense whose architecture combines a deterministic Lipschitz-constrained design—implemented via spectrally normalized convolutions—with two stochastic smoothing modules: (i) spectrally normalized random-projection filters and (ii) randomized attention-noise injection.* These components inject controlled smoothing noise, thereby incorporating stochasticity into the architecture and yielding an overall $\leq 2$-Lipschitz network that admits formal certification while consistently enhancing empirical robustness to strong $\ell_\infty$ attacks.

The key contributions of this paper can be summarized as follows:

1. **Hybrid defense.** We introduce `HyCAS`, *a randomized Lipschitz-constrained defense that provides both certified $\ell_2$ guarantees and strong empirical $\ell_\infty$ robustness across diverse vision benchmarks.*
2. **Theoretical guarantees.** We derive a tight $\ell_2$ robustness certificate for `HyCAS` and show that it remains competitive in empirical adversarial robustness against state-of-the-art $\ell_\infty$ attacks.
3. **Plug-and-play design.** `HyCAS` integrates a 1-Lipschitz deterministic core—implemented via spectrally normalized convolutions—with two stochastic modules: spectral normalized random-projection filters and randomized attention noise injection. These components inject controlled smoothing noise, thereby incorporating refined stochasticity into the network. Together they form a $\leq 2$-Lipschitz network that admits formal certification while boosting empirical robustness.
4. **Comprehensive evaluation.** Experiments on several benchmarks confirm that `HyCAS` outperforms prior certified and empirical defenses while allowing controllable trade-offs between certificate and empirical adversarial robustness.

## 2 RELATED WORK

**Deterministic certified defences.** Bounding a network's global Lipschitz constant makes its predictions provably stable to small $\ell_2$ perturbations. Early studies constrain fully–connected layers via spectral normalisation or orthogonal parameterisations (Miyato et al., 2018). Layer-wise Orthogonal Training (`LOT`) (Xu et al., 2022) and the Spectral–Lipschitz Lattice (`SLL`) (Araujo et al., 2023) extend these ideas to `CNN`s, yet often sacrifice clean accuracy on high-resolution data. Our deterministic backbone inherits its 1-Lipschitz guarantee while compensating for the accuracy drop through stochastic branches.

**Stochastic certified defences.** Randomised smoothing (`RS`) converts any base classifier into a Gaussian ensemble whose majority vote is certifiably robust (Cohen et al., 2019). Subsequent work enlarges certificates via adversarially trained bases (Salman et al., 2019), consistency regularisation (Jeong & Shin, 2020), and noise-aware denoising (Carlini et al., 2023). Mixing multiple noise scales further tightens guarantees, as shown by Dual RS (`DRS`) (Xia et al., 2024), Incremental RS (`IRS`) (Ugare et al.), and Adaptive RS (`ARS`) (Lyu et al., 2024). Our `HyCAS` departs from pure input–noise smoothing by injecting *internal* randomness via dual stochastic noise—yet still preserves a global $\leq 2$-Lipschitz certificate.

**Empirical defences.** Empirical methods drop certificates to maximise robustness against high-budget $\ell_\infty$ attacks. `PNI` (He et al., 2019) learns layer-wise Gaussian noise during adversarial training, boosting both clean and robust accuracy. `Learn2Perturb` (Jeddi et al., 2020) generalises this idea by jointly optimising feature-perturbation modules in an EM-like loop. `CTRW` (Ma et al., 2023) resamples convolution kernels at inference under learned mean–variance constraints, while `RPF` (Dong & Xu, 2023) freezes part of the first convolution layer as Gaussian projections, both outperforming strong PGD-trained baselines. In contrast, `CAP` (Xiang et al., 2023) infuses lung-

---

[1] In our experiments, we use natural vision and medical imaging datasets *as diverse imaging benchmarks*.

edge priors, bolstering adversarial robustness in `COVID-19 CT` prediction. *Despite these gains, these empirical defences provide no certified worst case guarantees (Yang et al., 2022; Liu et al., 2021), and many rely on input independent randomization (He et al., 2019; Jeddi et al., 2020; Dong & Xu, 2023); these non-certified randomized defences have often been circumvented by adaptive attacks that explicitly average over the internal noise (Athalye et al., 2018; Tramèr et al., 2020).*

Most prior defenses optimize for *either* certified *or* empirical robustness and are validated on a single regime—usually natural images, with only a few addressing specialised medical data. `HyCAS` *bridges this gap by incorporating a deterministic* 1-*Lipschitz architecture with stochastic smoothing modules (e.g., random-projection and attention-noise mechanisms), thereby forming a randomized defense that robustly generalizes across diverse imaging benchmarks.* A modest reduction in clean accuracy yields simultaneous performance gains in certified $\ell_2$ *and* empirical $\ell_\infty$ robustness (Fig. 4). Consequently, HyCAS aims to surpass the strongest deterministic certifiers *and* the leading empirical defences. Further details appear in Appendix A.1 (related work) and Appendix A.2 (preliminaries).

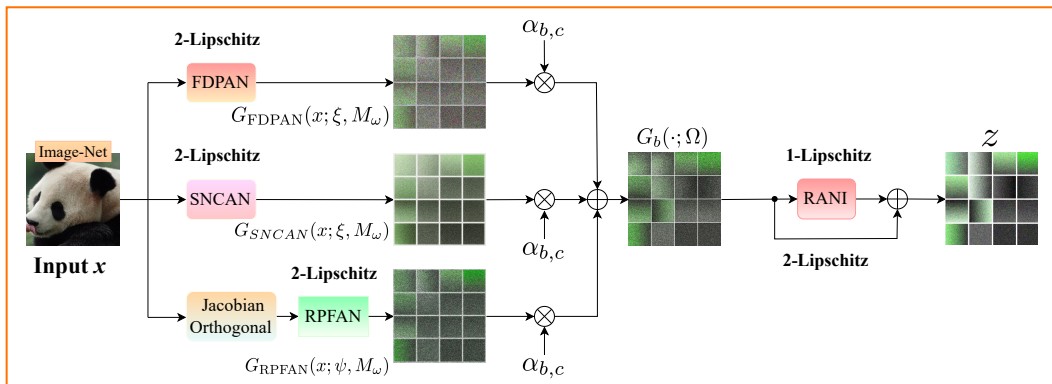

Figure 1: Overview of HyCAS mechanism. It consists of three parallel streams—FDPAN, SNCAN, and RPFAN—each built from 1-Lipschitz cores with Randomized Attention Noise Injection (RANI) residuals. Per-channel convex gating fuses the streams to form $G_b(;\Omega)$. Each stream is $\leq$ 2-Lipschitz; the fused stream and the stacked network remain $\leq$ 2-Lipschitz, enabling a margin-based $\ell_2$ certificate.

## 3 HYBRID CONVOLUTIONS WITH ATTENTION STOCHASTICITY

Randomized defenses often incorporate stochasticity into deep network structures by *(i)* tuning data-dependent hyper-parameters (e.g. noise scale, sampling rate) or *(ii) data-independent* architectural modifications. However, deep networks remain highly vulnerable on vision benchmarks, where imperceptible perturbations can sharply degrade accuracy. *Randomization alone is often insufficient; coupling it with a Lipschitz-constrained deterministic architecture yields stronger certified and empirical robustness.*

To address these limitations, we propose **Hybrid Convolutions with Attention Stochasticity (`HyCAS`)**, which replaces each convolutional layer in standard `CNN` backbones with Lipschitz-bounded stochastic streams that inject refined smoothing and controlled randomness into the network via two complementary, data-independent components—(i) a *Lipschitz-constrained deterministic architecture* and (ii) a *dual stochastic design*—thereby improving adversarial robustness.

Let $x \in \mathbb{R}^{H \times W \times C}$ be an input feature map with spatial dimensions $(H, W)$ and $C$ channels with label $y \in \mathcal{Y} = \{1, \dots, K\}$. We denote by $\|\cdot\|_2$ the Euclidean norm over vectorized tensors and by $\text{Lip}(h)$ the (global) $\ell_2$-Lipschitz constant of a map $h$ ("$L$-Lipschitz" means $\|h(u) - h(v)\|_2 \leq L\|u-v\|_2$). Our proposed HyCAS–integrated any base classifier $f_\theta$ with parameters $\theta$. The smoothed classifier induced by HyCAS is:

$$g_\theta(x) = \arg \max_{c \in \mathcal{Y}} \ \mathbb{P}_{\varepsilon,\Omega}\big[f_\theta(x + \varepsilon; \Omega) = c\big]. \tag{1}$$

where $\Omega = (\xi, \psi, M_\omega), \varepsilon \sim \mathcal{N}(0, \sigma^2 I)$ denotes Gaussian noise with mean 0 and standard deviation $\sigma$ matching the dimensions of $x$ to enable randomized smoothing, $\xi$ induces *deterministic Lipschitz-constrained structure*, $\psi$ integrates *implicit structural randomness* (first-level stochastic defense), and $M_\omega$ injects the *explicit attention noise* (second-level stochastic defense). The classifier $g_\theta(\cdot)$

returns whichever class $f_\theta$ is most likely to return, taking expectations over the distributions $\mathcal{N}(x + \varepsilon; x, \sigma^2 I)$. An overview is given in Fig. 1; *pseudocode is provided in App. A.7-A.8 (Algorithms 1–3)..*

`HyCAS` processes every feature map through three parallel streams: (a) *Frequency-aware Deterministic Projection with Attention Noise* (`FDPAN`); (b) *Spectrally Normalized Convolution with Attention Noise* (`SNCAN`); and (c) *Random Projection Filter with Attention Noise* (`RPFAN`). Their outputs are fused by a data-independent convex channel gate that down-weights high-sensitivity streams, thereby weakening naïve adversarial attacks.

Specifically, let $\mathcal{B} = \{\text{FDPAN}, \text{RPFAN}, \text{SNCAN}\}$ be the set of streams and for each stream $b \in \mathcal{B}$, let $G_b(\cdot; \Omega)$ denote its output feature map and those output feature maps are fused by channel-wise convex gate $\alpha_{b,c}$. For learnable, data-independent logits $\lambda_{b,c}$, we define channel-wise convex weights $\alpha_{b,c} = \frac{\exp(\lambda_{b,c})}{\sum_{b' \in \mathcal{B}} \exp(\lambda_{b',c})}$, such that $\sum_b \alpha_{b,c} = 1$, $\alpha_{b,c} \geq 0$, thereby we obtain the `HyCAS` block output is

$$z(x)_{:,:,c} = \sum_{b \in \mathcal{B}} \alpha_{b,c} \left[G_b(x; \Omega)\right]_{:,:,c} + R\left(\sum_{b \in \mathcal{B}} \alpha_{b,c} \left[G_b(x; \Omega)\right]_{:,:,c}; M_\omega\right); \quad c = 1, \ldots, C. \quad (2)$$

where $R$ denotes RANI module.

**Convex fusion and expected-logit map are $\leq$ 2-Lipschitz.** If each stream satisfies $\text{Lip}(G_b) \leq 2$, and the gate is convex (Eq. 2), then the per-channel fusion has $\text{Lip}(x \mapsto z(x)) \leq \max_{b \in B} \text{Lip}(G_b) \leq 2$. (*ref. Appendix A.3 (Prop. 4)*). Taking the expectation over $\Omega$ preserves the Lipschitz constant (*ref. Appendix A.3 (Lemma 2)*), so the network's expected logit map $Z(x) := \mathbb{E}_\Omega\left[s_\theta(x; \Omega)\right]$ remains $\leq$ 2-Lipschitz. Formally:

**Theorem 1** (HyCAS block is $\leq$ 2-Lipschitz). *Each constituent stream—SNCAN, RPFAN, and FDPAN with skip weight $\beta \leq \frac{1}{3}$—is individually $\leq$ 2-Lipschitz. Indeed, every stream is the composition of three maps: (i) a stochastic projection $\mathcal{T}_\psi$ (seeded by $\psi$), 1-Lipschitz; (ii) a deterministic projection $\mathcal{D}_\xi$ (parameterized by $\xi$), 1-Lipschitz; and (iii) a stochastic attention noise $M_\omega \colon \mathbb{R}^d \to [0,1]^d$, 1-Lipschitz. For any input $x$, the resulting feature map*

$$G_b(x; \xi, \psi, \omega) = \mathcal{D}_\xi(\mathcal{T}_\psi(x)) + M_\omega(\mathcal{D}_\xi(\mathcal{T}_\psi(x)))$$

*is therefore 2-Lipschitz on each forward pass. The subsequent per-channel convex fusion is non-expansive, so it cannot increase the Lipschitz constant. Consequently, every HyCAS block is provably $\leq$ 2-Lipschitz (see Appendix A.3 (Prop. 4)).*

*Proof.* See proof within the Appendix A.3. $\square$

**Corollary 1** (*Randomized Lipschitz margin certificate for expected logits*). *Let $Z(x)$ be HyCAS logits averaged over internal randomness, with $\text{Lip}(Z) \leq 2$. Let $\Delta(x) = Z_{(1)}(x) - Z_{(2)}(x)$ is the gap between the top-two logits. Then $r_2(x) = \frac{\Delta(x)}{4}$ is is a valid $\ell_2$ certificate: for all $\|\delta\|_2 < r_2(x)$, we have $\arg\max Z(x + \delta) = \arg\max Z(x)$. This is the HyCAS pointwise $\ell_2$ certificate (App. A.4).*

*Proof.* See proof within the Appendix A.3. $\square$

The `HyCAS`–integrated network is optimised with a standard $\ell_2$ loss as:

$$\mathcal{L}_{HyCAS} = \zeta \odot \mathcal{L}_{FDPAN} + \varphi \odot \mathcal{L}_{SNCAN} + \nu \odot \mathcal{L}_{RPFAN} + \kappa \odot \mathcal{L}_{RANI}, \quad (3)$$

where $\zeta$, $\varphi$, $\nu$, and $\kappa$ denoted by learnable parameters, while $\odot$ represents Hadamard product.

All streams are spectrally normalised ($\|W\|_2 \leq 1$) and the stochastic attention noise module is 1-Lipschitz. Hence, by Theorem 1–Corollary 1, every `HyCAS` block—and any network built by stacking them—is $\leq$ 2-Lipschitz, so attacks with $\ell_2$-norm $< \Delta(x)/4$ cannot alter the prediction.

## 3.1 FREQUENCY-AWARE DETERMINISTIC PROJECTION WITH ATTENTION NOISE (FDPAN)

Under $\ell_2$-bounded attacks, adversaries (i) conceal perturbations in high-frequency `DCT` coefficients and (ii) exploit channel-wise gradient regularities that generalize across models. `FDPAN` counters both phenomena by weaving *frequency truncation*, *channel scrambling*, *spectral control*, and *calibrated stochasticity* into the architecture.

`FDPAN` is a four-stage cascade (*see Appendices A.3–A.5 for Lemma 3 and Figure 5*), where each component comprises a deterministic 1-Lipschitz core followed by two randomized residuals, i.e.,

Table 1: Certified accuracy (%) of `HyCAS` and prior baselines on `CIFAR-10` and `ImageNet`. Bold value denotes the best in each column across all noise–radius pairs. All methods are evaluated at two noise levels.

| Approaches | $\sigma$ | CIFAR-10 | | | | | | | | ImageNet | | | | | | | |
|---|---|---|---|---|---|---|---|---|---|---|---|---|---|---|---|---|---|
| | | Certified accuracy at predetermined $\ell_2$ radius $r$ (%) | | | | | | | | Certified accuracy at predetermined $\ell_2$ radius $r$ (%) | | | | | | | |
| | | 0.00 | 0.25 | 0.50 | 0.75 | 1.00 | 1.25 | 1.50 | 2.0 | 0.00 | 0.25 | 0.50 | 0.75 | 1.00 | 1.25 | 1.50 | 2.0 |
| RS | 0.25 | 75.3 | 60.2 | 43.4 | 26.1 | 0 | 0 | 0 | 0 | 67.1 | 48.7 | 0 | 0 | 0 | 0 | 0 | 0 |
| | 0.50 | 65.2 | 54.1 | 41.3 | 32.4 | 23.2 | 14.7 | 9.34 | 0 | 57.3 | 45.9 | 36.8 | 28.7 | 0 | 0 | 0 | 0 |
| IRS | 0.25 | 78.6 | 63.2 | 47.5 | 30.8 | 19.6 | 10.3 | 5.72 | 0 | 68.4 | 58.5 | 46.2 | 38.7 | 32.1 | 19.3 | 10.8 | 0 |
| | 0.50 | 71.3 | 58.5 | 44.1 | 33.3 | 24.1 | 15.7 | 11.4 | 2.2 | 62.4 | 50.9 | 41.5 | 34.7 | 27.3 | 20.2 | 13.8 | 6.31 |
| DRS | 0.25 | 83.4 | 65.8 | 50.2 | 34.5 | 24.7 | 15.8 | 10.5 | 0 | 70.6 | 61.2 | 51.8 | 42.7 | 38.4 | 32.6 | 25.4 | 0 |
| | 0.50 | 78.1 | 62.1 | 48.7 | 35.8 | 24.5 | 17.9 | 12.9 | 4.6 | 67.6 | 58.2 | 49.6 | 42.8 | 35.6 | 33.2 | 29.8 | 21.3 |
| ARS | 0.25 | 84.1 | 67.3 | 51.4 | 39.1 | 30.9 | 21.1 | 16.2 | 0 | 71.1 | 61.4 | 52.7 | 43.1 | 39.1 | 33.4 | 26.7 | 0 |
| | 0.50 | 78.4 | 63.7 | 50.2 | 38.9 | 31.8 | 23.3 | 19.7 | 8.47 | 68.1 | 58.7 | 50.3 | 43.4 | 39.1 | 34.5 | 30.6 | 22.4 |
| LOT | — | 80.5 | 64.7 | 48.6 | 34.3 | 23.6 | 15.2 | 9.14 | 0 | 69.7 | 60.6 | 50.9 | 42.2 | 37.1 | 30.5 | 21.8 | 0 |
| | | 76.7 | 60.4 | 46.3 | 35.1 | 24.9 | 17.3 | 12.1 | 6.25 | 66.1 | 57.4 | 48.9 | 42.8 | 38.4 | 32.9 | 28.3 | 19.8 |
| SLL | — | 81.4 | 65.3 | 49.9 | 33.1 | 23.6 | 14.7 | 9.94 | 0 | 70.2 | 57.7 | 48.4 | 41.8 | 37.6 | 31.9 | 24.3 | 0 |
| | | 77.9 | 62.6 | 48.7 | 34.5 | 24.4 | 16.2 | 13.7 | 5.83 | 67.3 | 55.5 | 49.1 | 42.8 | 39.1 | 34.5 | 26.7 | 21.3 |
| HyCAS | 0.25 | **85.4** | **70.1** | **56.7** | **44.3** | **36.5** | **29.6** | **22.9** | **8.52** | **72.3** | **63.9** | **55.6** | **46.4** | **40.7** | **35.2** | **29.7** | **5.42** |
| | 0.50 | **80.7** | **65.3** | **54.8** | **44.3** | **36.8** | **30.3** | **23.4** | **12.5** | **69.2** | **60.6** | **53.9** | **45.6** | **41.1** | **36.3** | **32.7** | **24.8** |

2-Lipschitz: (i) *Low-pass `DCT` mask* (1-Lipschitz) — excises fragile high-frequency bands. (ii) *Orthogonal Jacobian $1 \times 1$ matrix + Randomized Attention Noise Injection (`RANI`)* (2-Lipschitz) — scrambles channel gradients and injects structured noise. (iii) `SNCAN` (2-Lipschitz) — keeps the convolutional kernel spectrum bounded while introducing additional stochasticity. (iv) `RANI` that further incorporates refined randomness and is 2-Lipschitz.

Let the deterministic core be $H(x) = C_{K_e}\left(U\,\Phi^\top(\Lambda \odot \Phi x)\right)$, where $C_{K_e}$ be Spectrally normalized convolution, $U$ denotes Orthogonal Jacobian matrix layer, $\Phi$ is the orthonormal 2-D `DCT` and $\Lambda$ is the low-pass mask.

To incorporate richer, refined stochasticity, we apply `RANI` immediately after the deterministic core. Given an attention noise $M_\omega$ and a noise–strength parameter $\omega \in [0,1]$, a `RANI` module is denoted by $R(x; M_\omega)$, and is 1–Lipschitz for every freshly drawn $\omega$ during both training and inference. Hence, combining the deterministic path and two independent `RANI` injections (stochastic) yields the `FDPAN` stream output as: $G_{\text{FDPAN}}(x; \xi, M_\omega) = H(x;\xi) + R\left(H(x;\xi); M_{\omega_i}\right)$, where $H(;\xi)$ is the deterministic core, and each $R\left(;\xi, M_{\omega_i}\right)$ introduces independent stochastic attention noise $M_{\omega_i}$ for $i \in \{1:2\}$. Because the two $M_{\omega_i}$ terms are cascaded $i$ and the entire stream is at most 2-Lipschitz by the triangle inequality. The skip connection is 1-Lipschitz as well. *Notably, more details about `SNCAN` module and `RANI` mechanism are in the following sections.* Formally:

**Proposition 1** (`FDPAN` is at most 2-Lipschitz)**.** *Assume the deterministic core $H(\cdot)$ is 1-Lipschitz and that, for every attention noise $M_{\omega_i}$, the `RANI` $R(\cdot; \omega_i)$ is also 1-Lipschitz; the skip connection is likewise 1-Lipschitz. Define the `FDPAN` stream output by $G_{\text{FDPAN}}(x; M_\omega) = H(\cdot) + (R(\cdot) \circ H(\cdot))$ is 4-Lipschitz and therefore satisfies* $\text{Lipschitz}(G_{\text{FDPAN}}) \leq 2$.

*Proof.* See proof within the Appendix A.3. □

Therefore, `FDPAN` minimises the objective of `HyCAS` by incorporating refined stochasticity into the network through all employed modules as mentioned in the above:

$$\mathcal{L}_{\text{FDPAN}}(\theta) = \min_\theta \; \mathbb{E}_{(x,y)} \, \mathbb{E}_{\varepsilon \sim \mathcal{N}(0, \sigma^2 I), \, \xi, M_{\omega_i}} \; \ell\big(f_\theta(x + \varepsilon; \xi, M_{\omega_i}), y\big). \tag{4}$$

## 3.2 SPECTRALLY NORMALIZED CONVOLUTIONS WITH ATTENTION NOISE (SNCAN)

To design `SNCAN` (*see Appendix A.5 (Figure 6)*), we replace every standard convolutional layer with a *spectrally normalized convolution* (`SNC`; see Appendix A.2.2). This substitution introduces controlled gradient variability while preserving the deterministic 1-Lipschitz bound on worst-case $\ell_2$ perturbations. However, the resulting *stationary* gradient fields can still be exploited by adversarial attacks. To mitigate this vulnerability and further strengthen robustness, we incorporate our *data-independent* `RANI` module (Section 3.4) to each `SNC` layer, thereby injecting fine-grained stochasticity while preserving a tight Lipschitz envelope.

Let $v = C_{K_e}(x)$, $R(v; M_\omega) = D_\omega v$, where $C_{K_e}$ is an SNC with kernel $K_e$ rescaled to satisfy $\|K_e\|_{\mathrm{op}} \leq 1$ and $D_\omega = \mathrm{diag}(M_\omega)$ is a diagonal matrix formed from the attention-noise tensor $M_\omega \in [0, 1]^{H \times W \times C}$. Because every diagonal entry of $D_\omega$ lies in $[0, 1]$, we have $\|D_\omega\|_2 \leq 1$. Hence $\mathrm{Lip}(C_{K_e}) \leq 1$, $\mathrm{Lip}(R(\cdot; M_\omega)) \leq 1$, $\mathrm{Lip}(I + D_\omega) \leq 2$.

`RANI` generates a bounded attention noise $M_\omega \in [0, 1]^{H \times W \times C}$ and forms the stochastic residual output as:

$$G_{SNCAN}(x; \xi, M_\omega) = C_{K_e}(x; \xi) + R(C_{K_e}(x; \xi); M_\omega) = (I + D_\omega) C_{K_e}(x; \xi). \tag{5}$$

By incorporating `RANI` into a deterministic, 1-Lipschitz convolutional block, we obtain a *randomized* defense that is provably 2-Lipschitz, as formalized below.

**Proposition 2** (2-Lipschitz hybrid block)**.** *For every input pair $x, y \in \mathbb{R}^{H \times W \times C}$ and every noise sample $M_\omega$,*

$$\|G_{\mathrm{SNCAN}}(x; M_\omega) - G_{\mathrm{SNCAN}}(y; M_\omega)\|_2 \ \leq \ 2 \|x - y\|_2.$$

*Proof.* See proof within the Appendix A.3. $\qquad\square$

Each `SNCAN` block, therefore, multiplies the network's global Lipschitz constant by at most 2 while injecting fresh randomness on every forward pass, synchronizing the gradient landscape that an adversary sees. Therefore, `SNCAN` minimizes the objective of `HyCAS` by incorporating refined stochasticity into the network through `SNC` and `RANI` modules:

$$\mathcal{L}_{\mathrm{SNCAN}}(\theta) = \min_\theta \ \mathbb{E}_{(x,y)} \ \mathbb{E}_{\varepsilon \sim \mathcal{N}(0, \sigma^2 I), \ \xi, M_\omega} \ \ell(f_\theta(x + \varepsilon; \xi, M_\omega), y). \tag{6}$$

*In summary, spectral normalization (e.g., `SNC`) complemented by `RANI` yields a randomized module built on a deterministic architecture, whose Lipschitz envelope remains tight while its gradients vary across evaluations.*

### 3.3 RANDOM-PROJECTION CONVOLUTION WITH ATTENTION NOISE (RPFAN)

`RPFAN` couples a spectrally controlled random projection (1-Lipschitz) with a data-*independent* randomized attention residual (2-Lipschitz). It therefore introduces *dual* stochasticity—(i) from the random projection itself and (ii) from `RANI`—while keeping the stream's Lipschitz constant at most 2. In practice, both the random projection and attention noise are freshly resampled as described in §3.5. *Network Execution*.

The `RPFAN` module (*see Appendix A.5 (Figure 7)*) inherits the Johnson–Lindenstrauss (`JL`) embedding guarantee of a *random-projection filter* (`RPF`) (Dong & Xu, 2023) (see Appendix A.2.3 for details) and extends it with three carefully chosen components: (i) two **core innovations** that render the module *certifiably 1-Lipschitz*, and (ii) the `RANI` module, which raises the Lipschitz constant to 2 while injecting an additional source of *data-independent* stochasticity. Combined, these elements provide *dual stochasticity*—one arising from the random projection itself and the other from `RANI`—thereby strengthening adversarial robustness without exceeding a 2-Lipschitz bound. The three components are summarized below.

1. **Energy-preserving channel pre-mix.** Before the random-projection filter is applied, we leverage 1×1 orthogonal Jacobian matrix as channel mixer $U$ with $U^\top U = I$ (Horn & Johnson, 2012) to apply $x \mapsto Ux$, which equalises channel energy so that every spatial dimension enters the projection space with identical energy distribution (*see Appendix A.3 (Lemma 3)*).
2. **Batch-aware spectral normalisation for random projection.** The random-projection filter $W_0$ is sampled exactly as in (Dong & Xu, 2023) (ref. A.2.3). We then rescale it using a *per-sample*, two-step power-iteration (PI) scheme: (i) Draw $u \sim \mathcal{N}(0, 1)$ of shape $(N, \frac{H}{s}, \frac{W}{s}, C_{\mathrm{out}})$; (ii) Update twice $v \leftarrow \frac{\mathrm{Conv}^\top(u; W_0)}{\|\cdot\|_2}$, $u \leftarrow \frac{\mathrm{Conv}(v; W_0)}{\|\cdot\|_2}$, *normalising each sample independently*; and (iii) compute the Rayleigh quotient (`RQ`), thereby to form a spectral normalized random projection filter $W_{\mathrm{SN}}$ as:

$$RQ = \frac{1}{N} \sum_n \langle u_n, \mathrm{Conv}(v_n; W_0) \rangle, \quad W_{\mathrm{SN}} = \frac{W_0}{\max(RQ, 1)}.$$

This *batch-aware* `PI` yields a tighter bound on $\|\mathrm{Conv}(\cdot; W_0)\|_2$ than layer-wise PI while guaranteeing that the projection remains 1-Lipschitz (Appendix A.3).

Table 2: Certified accuracy (%) of `HyCAS` and prior defenses on `CelebA`, `HAM10000`, and `NIH-CXR`. Boldface denotes the best in each column across all noise–radius pairs. Methods are evaluated at 3 noise levels.

| Approaches | $\sigma$ | CelebA $\ell_2$ radius $r$ (%) | | | HAM10000 $\ell_2$ radius $r$ (%) | | | NIH-CXR $\ell_2$ radius $r$ (%) | | |
|---|---|---|---|---|---|---|---|---|---|---|
| | | 0.0 | 0.50 | 1.0 | 0.0 | 0.50 | 1.0 | 0.0 | 0.50 | 1.0 |
| RS | 0.25 | 92.8 | 45.7 | 0 | 94.6 | 53.2 | 10.5 | 77.4 | 43.5 | 15.7 |
| | 0.50 | 87.7 | 47.8 | 10.5 | 89.3 | 52.1 | 12.2 | 73.3 | 39.9 | 21.8 |
| | 1.0 | 81.4 | 51.6 | 18.8 | 84.7 | 54.3 | 21.2 | 66.4 | 42.9 | 22.8 |
| ARS | 0.25 | 95.2 | 53.3 | 27.4 | 96.7 | 57.4 | 31.3 | 79.1 | 58.4 | 32.5 |
| | 0.50 | 91.3 | 53.9 | 30.4 | 91.9 | 55.1 | 32.8 | 74.9 | 54.7 | 33.3 |
| | 1.0 | 85.3 | 59.2 | 31.6 | 86.9 | 57.4 | 34.6 | 69.9 | 52.9 | 34.1 |
| HyCAS | 0.25 | **96.8** | **58.1** | **33.7** | **97.2** | **60.5** | **35.4** | **81.6** | **61.9** | **38.6** |
| | 0.50 | **92.7** | **59.3** | **34.8** | **93.1** | **60.4** | **36.6** | **76.2** | **58.6** | **40.9** |
| | 1.0 | **87.7** | **62.3** | **36.9** | **88.2** | **61.9** | **38.5** | **71.7** | **60.6** | **41.4** |

3. **Randomised Attention Noise Injection (`RANI`).** Given the 1-Lipschitz projection output $h = \mathrm{Conv}(Ux; W_{\mathrm{SN}})$, draw internal randomness $\omega$ and apply a *data-independent* bounded mask $M_\omega \in [0,1]^d$ through the `RANI` module $R(h; M_\omega)$. For newly drawn $\omega$, $\|I + D_{M_\omega}\|_2 \le \|I\|_2 + \|D_{M_\omega}\|_2 \le 2$; therefore the composite map $x \mapsto R\big(\mathrm{Conv}(Ux; W_{\mathrm{SN}}); \omega\big)$ is 2-Lipschitz. This couples the spectrally normalised random projection with `RANI`, injecting refined stochasticity while multiplying the stream's Lipschitz constant by at most 2 (ref. Proposition 2; see also App. A.2 for the residual bound). Define the 1-Lip core $H_{\mathrm{RPFAN}}(x) = \mathrm{Conv}(Ux; W_{\mathrm{SN}})$ and the stream output as:

$$G_{\mathrm{RPFAN}}(x; \psi, M_\omega) = H_{\mathrm{RPFAN}}(x; \psi) + R\big(H_{\mathrm{RPFAN}}(x; \psi); M_\omega\big); \tag{7}$$

then $\mathrm{Lip}\big(G_{\mathrm{RPFAN}}\big) \le 2$ (See Proof 5).

Therefore, `RPFAN` minimises the objective of `HyCAS` by incorporating refined stochasticity into the network through `RPF` and `RANI` modules:

$$\mathcal{L}_{\mathrm{RPFAN}}(\theta) = \min_\theta \ \mathbb{E}_{(x,y)} \ \mathbb{E}_{\varepsilon \sim \mathcal{N}(0,\sigma^2 I), \ \Omega} \ \ell\big(f_\theta(x + \varepsilon; \Omega), y\big). \tag{8}$$

### 3.4 RANI: RANDOMIZED ATTENTION NOISE INJECTION

**Motivation.** Certified deterministic 1-Lipschitz defenses (e.g., `SNC`) bound the worst–case $\ell_2$ perturbation but still expose a deterministic gradient field that adversaries can exploit. Classical randomized defenses inject noise only at the input, whereas certified Lipschitz defenses remain deterministic inside the network. **`RANI`** closes this gap: it injects a *data-independent*, stochastic *attention mask* $M_\omega \in [0,1]^d$ after every spectrally-normalised block in the three streams (`FDPAN`, `RPFAN`, `SNCAN`) and once more at their fused output, while preserving a global 2-Lipschitz envelope. Formally, the deterministic 1-Lipschitz map $h \in H(x; \xi, \psi)$ is replaced by the stochastic 2-Lipschitz map $\hat{h} \in R(h; M_\omega)$ via incorporating `RANI` module ($R(; M_\omega)$).

**Attention noise mechanism.** For each forward pass, `RANI` draws fresh noise $\omega \sim \mathcal{N}(0, I)$ for internal randomness and computes a bounded attention noise $M_\omega \in [0,1]^d$ that is *independent of the current features*. For any deterministic feature tensor $h \in \mathbb{R}^{H \times W \times C}$, we modulate it according to:

$$\hat{h} = h \odot M_\omega, \tag{9}$$

where $\odot$ denotes the Hadamard product. This yields a Lipschitz constant of at most 2; hence every block's constant grows from $1 - Lipschitz$ deterministic to a *randomised defense (see Appendix A.3 (Lemma 3))*.

In practice, $M_\omega$ is produced by our `RANI` module via injecting noise at local and channel information of the given deterministic feature maps (e.g., $h \in H(x; \xi, \psi)$); (See Appendix A.6 for more details of our `RANI` module. The following lemma states the guarantee formally.

**Lemma 1** (`RANI` module is 2-Lipschitz). *Let $h \in \mathbb{R}^d$ and let $M(\omega) \in [0,1]^d$ be sampled i.i.d. from an arbitrary distribution that is independent of $h$. Define the `RANI` mapping as shown in Eq. 9. Then, for each randomly drawn $\omega$ and any $h_1, h_2$, the mapping $\hat{h}$ is 2-Lipschitz with respect to the Euclidean norm $\| \cdot \|_2$; i.e.,*

$$\|R(h_1; M_\omega) - R(h_2; M_\omega)\|_2 \ \le \ 2 \, \|h_1 - h_2\|_2.$$

*Proof.* See proof within the Appendix A.3. $\square$

Table 3: Robust accuracy (%) against $\ell_\infty$ attacks (APGD-20 and AA-20) on `NIH-CXR` (left) and `NCT-CRC-HE-100K` (right) at $\epsilon \in \{8/255, 16/255\}$.

| Method | NIH-CXR | | | | | NCT-CRC-HE-100K | | | | |
|---|---|---|---|---|---|---|---|---|---|---|
| | Clean | APGD–20 | | AA–20 | | Clean | APGD–20 | | AA–20 | |
| | | 8/255 | 16/255 | 8/255 | 16/255 | | 8/255 | 16/255 | 8/255 | 16/255 |
| AT | $89.1 \pm 1.91$ | $74.7 \pm 2.52$ | $66.9 \pm 3.41$ | $74.2 \pm 2.93$ | $64.1 \pm 3.70$ | $92.2 \pm 1.82$ | $77.8 \pm 2.51$ | $68.7 \pm 3.12$ | $76.3 \pm 2.83$ | $66.2 \pm 3.61$ |
| RPF | $88.4 \pm 1.82$ | $83.7 \pm 2.49$ | $71.9 \pm 3.29$ | $82.5 \pm 2.71$ | $70.8 \pm 3.52$ | $91.1 \pm 1.71$ | $86.1 \pm 2.33$ | $73.9 \pm 3.33$ | $84.2 \pm 2.62$ | $72.4 \pm 3.41$ |
| CTRW | $88.4 \pm 1.73$ | $85.1 \pm 2.23$ | $73.1 \pm 3.22$ | $84.5 \pm 2.48$ | $72.6 \pm 3.41$ | $90.4 \pm 1.62$ | $87.6 \pm 2.29$ | $76.7 \pm 3.12$ | $86.7 \pm 2.44$ | $75.2 \pm 3.22$ |
| DCS | $87.2 \pm 2.05$ | $82.4 \pm 2.41$ | $71.7 \pm 3.21$ | $81.7 \pm 2.72$ | $69.6 \pm 3.45$ | $90.3 \pm 1.93$ | $84.5 \pm 2.72$ | $73.0 \pm 3.25$ | $83.3 \pm 2.74$ | $71.6 \pm 3.46$ |
| ARS | $84.8 \pm 2.22$ | $75.1 \pm 3.01$ | $64.7 \pm 3.28$ | $72.8 \pm 3.11$ | $62.8 \pm 3.72$ | $86.8 \pm 2.14$ | $75.9 \pm 2.71$ | $66.1 \pm 3.52$ | $74.6 \pm 3.11$ | $64.5 \pm 3.73$ |
| DRS | $83.9 \pm 2.33$ | $73.1 \pm 2.41$ | $62.9 \pm 3.23$ | $71.6 \pm 3.12$ | $61.9 \pm 3.81$ | $85.9 \pm 2.25$ | $75.1 \pm 2.61$ | $65.2 \pm 3.82$ | $73.5 \pm 3.21$ | $63.7 \pm 3.94$ |
| **HyCAS** | $\mathbf{89.5 \pm 1.64}$ | $\mathbf{88.6 \pm 2.33}$ | $\mathbf{77.3 \pm 3.14}$ | $\mathbf{86.9 \pm 2.42}$ | $\mathbf{74.4 \pm 3.33}$ | $\mathbf{91.3 \pm 2.63}$ | $\mathbf{90.4 \pm 2.82}$ | $\mathbf{79.3 \pm 3.52}$ | $\mathbf{88.2 \pm 2.63}$ | $\mathbf{76.7 \pm 3.34}$ |

Table 4: Robust accuracy (%) against $\ell_\infty$ attacks (APGD-20 and AA-20) on `HAM10000` (left) and `EyePACS` (right) at $\epsilon \in \{8/255, 16/255\}$.

| Method | HAM10000 | | | | | EyePACS | | | | |
|---|---|---|---|---|---|---|---|---|---|---|
| | Clean | APGD-20 | | AA–20 | | Clean | APGD-20 | | AA–20 | |
| | | 8/255 | 16/255 | 8/255 | 16/255 | | 8/255 | 16/255 | 8/255 | 16/255 |
| AT | $75.2 \pm 2.94$ | $56.1 \pm 3.49$ | $46.5 \pm 3.75$ | $54.2 \pm 3.93$ | $44.2 \pm 3.80$ | $78.2 \pm 2.91$ | $60.0 \pm 2.72$ | $50.1 \pm 3.52$ | $58.3 \pm 2.83$ | $48.2 \pm 3.61$ |
| RPF | $74.3 \pm 2.86$ | $64.1 \pm 3.41$ | $51.9 \pm 3.42$ | $62.6 \pm 3.71$ | $50.4 \pm 3.58$ | $77.1 \pm 2.90$ | $67.8 \pm 2.57$ | $56.1 \pm 3.12$ | $66.4 \pm 2.73$ | $54.4 \pm 3.44$ |
| CTRW | $74.3 \pm 2.75$ | $64.7 \pm 3.33$ | $52.8 \pm 3.32$ | $63.3 \pm 3.48$ | $51.2 \pm 3.52$ | $76.4 \pm 2.84$ | $70.1 \pm 2.53$ | $57.7 \pm 3.35$ | $69.7 \pm 2.64$ | $56.1 \pm 3.31$ |
| DCS | $73.2 \pm 2.94$ | $62.9 \pm 3.48$ | $51.7 \pm 3.09$ | $61.4 \pm 3.72$ | $49.5 \pm 3.45$ | $76.4 \pm 2.94$ | $66.8 \pm 2.53$ | $55.2 \pm 3.68$ | $65.3 \pm 2.74$ | $53.6 \pm 3.46$ |
| ARS | $69.8 \pm 3.22$ | $53.9 \pm 3.88$ | $44.1 \pm 3.13$ | $52.7 \pm 4.10$ | $42.8 \pm 3.71$ | $72.9 \pm 3.97$ | $59.9 \pm 2.91$ | $48.8 \pm 3.61$ | $57.6 \pm 2.94$ | $46.5 \pm 3.73$ |
| DRS | $68.9 \pm 3.28$ | $53.4 \pm 3.84$ | $43.2 \pm 3.43$ | $51.6 \pm 4.12$ | $41.8 \pm 3.81$ | $71.9 \pm 3.86$ | $58.3 \pm 2.61$ | $47.4 \pm 3.71$ | $56.5 \pm 2.92$ | $45.7 \pm 3.94$ |
| **HyCAS** | $\mathbf{74.6 \pm 2.74}$ | $\mathbf{67.8 \pm 3.43}$ | $\mathbf{55.3 \pm 3.14}$ | $\mathbf{65.8 \pm 3.42}$ | $\mathbf{53.1 \pm 3.33}$ | $\mathbf{77.6 \pm 2.79}$ | $\mathbf{72.6 \pm 2.72}$ | $\mathbf{60.5 \pm 3.43}$ | $\mathbf{71.8 \pm 2.82}$ | $\mathbf{58.3 \pm 3.32}$ |

Therefore, `RANI` minimises the objective of `HyCAS` by incorporating refined stochasticity into the network:

$$\mathcal{L}_{\text{RANI}}(\theta) = \min_\theta \, \mathbb{E}_{(x,y)} \, \mathbb{E}_{\varepsilon \sim \mathcal{N}(0,\sigma^2 I), \, M_\omega} \, \ell\big(f_\theta(x + \varepsilon; M_\omega), y\big), \tag{10}$$

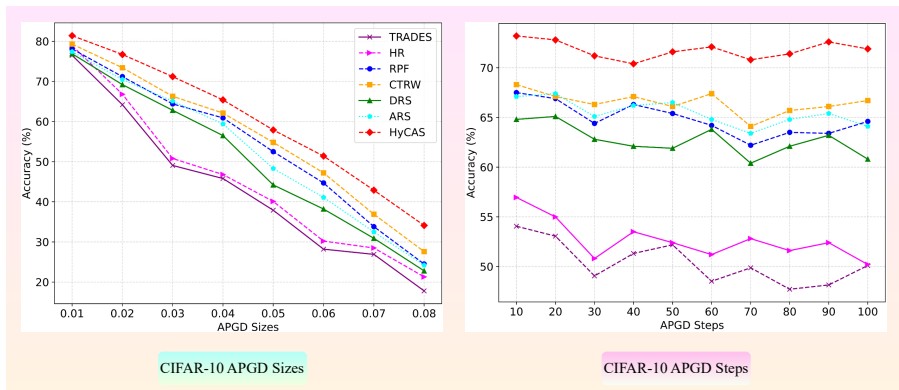

Figure 2: Empirical robustness of `HyCAS` versus leading baselines (`RPF`, `CTRW`, `DRS`, `ARS`) on `CIFAR-10` under strong `APGD` attacks. We evaluate two settings: (1) perturbation sizes $\epsilon$ from 0.01 to 0.08 and (2) iteration steps from 10 to 100.

## 4 EXPERIMENT RESULT

### 4.1 EXPERIMENT SETUP

**Evaluation protocol.** We evaluate `HyCAS` on eight vision benchmarks (`CIFAR-10/100` (Krizhevsky, 2009), `ImageNet-1k` (Deng et al., 2009), `CelebA` (Liu et al., 2015), `NCT-CRC-HE-100K` (Kather et al., 2018), `NIH-CXR` (Wang et al., 2017), `EyePACS` (EyePACS, 2015), and `HAM10000` (Tschandl et al., 2018)). We report *certified accuracy* at preset $\ell_2$ radii $r$ for smoothing noise levels $\sigma \in \{0.25, 0.50, 1.0, 2.0\}$.

Empirical robustness is measured under $\ell_\infty$ `APGD-20` *(Croce & Hein, 2020)* [2] and AutoAttack (AA) (Croce & Hein, 2020) at budgets $\epsilon \in \{8/255, 16/255\}$. We also evaluate `HyCAS` under

---

[2] We use the combination of $\ell_\infty\text{-APGD}_{CE}$ and $\ell_\infty\text{-APGD}_{T-DLR}$ from (Croce & Hein, 2020), each run for 20 iterations with 5 random restarts; we denote this union as `APGD-20`.

stronger `APGD` settings with larger $\epsilon$ and more attack steps (Figs. 2–4). Baselines span randomized smoothing methods (`RS` (Cohen et al., 2019), `IRS`(Ugare et al.), `DRS` (Xia et al., 2024)), `ARS` (Lyu et al., 2024), and 1-Lipschitz defenses (`LOT` (Xu et al., 2022), `SLL` (Araujo et al., 2023)). *All experiments are run with five random seeds, and we report the mean, the standard deviation, or both for each experiment.* Implementation details, certification and empirical settings are in Appendix A.8.

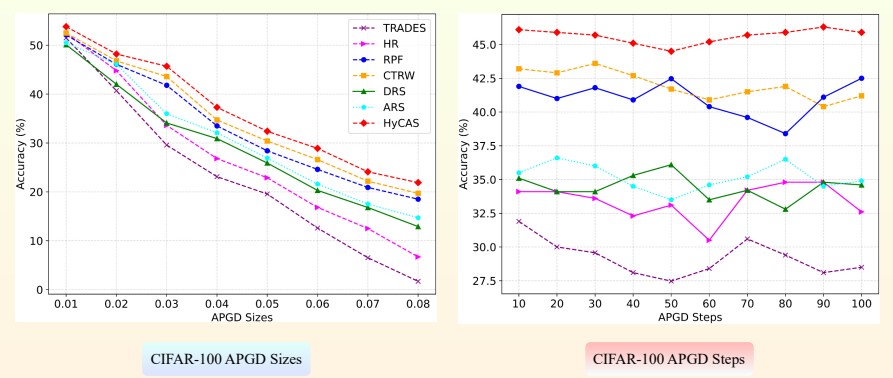

Figure 3: Empirical robustness of `HyCAS` versus leading baselines (`RPF`, `CTRW`, `DRS`, `ARS`) on `CIFAR-100` under strong `APGD` attacks. We evaluate two settings: (1) perturbation sizes $\epsilon$ from 0.01 to 0.08 and (2) iteration steps from 10 to 100.

## 4.2 Certified Adversarial Robustness at Preset Radii

**CIFAR-10 and ImageNet.** Across all baselines in Table 1, `HyCAS` achieves the best certified accuracy for every $(r, \sigma)$ pair. On the `CIFAR-10`, at the representative medium radius $r=0.75$, it yields **44.3%** certified accuracy for both $\sigma \in \{0.25, 0.50\}$—an gain of 5.2–18.2% over the prior methods. Even in the large-radius tail ($r=2.0$, $\sigma = 0.50$), it retains **12.5%**, surpassing the leading baseline by 4.0–12.5%. A similar trend emerges on `ImageNet`: in the large-radius regime ($r=1.5$, $\sigma = 0.50$), `HyCAS` reaches **32.7%** certified accuracy, exceeding every baseline by 2.1–32.7%. `HyCAS` also delivers state-of-the-art clean accuracy—85.4% on `CIFAR-10` and 72.3% on `ImageNet`—modestly but consistently ahead of all baselines.

**Skin, Chest Xray, and Face datasets.** Table 2 demonstrates the same dominance beyond CIFAR-10 and ImageNet datasets. On the `CelebA` dataset, `HyCAS` achieves certified accuracies of **62.3%** at $r = 0.5$ and **36.9%** at $r = 1.0$, outperforming RS and ARS by 5.3–18.1%. For the `HAM10000` dataset, it reaches **61.9%** (at $r = 0.5$) and **38.5%** (at $r = 1.0$), leading all baselines by approximately 4%. On the `NIH-CXR` dataset, certified accuracy spans **61.9%** (at $r = 0.5$, $\sigma = 0.25$) to **41.4%** (at $r = 1.0$, $\sigma = 1.0$), a gain of 3.5–7.3% over the leading baseline (e.g., `ARS`). Clean accuracy is likewise higher or on par across the board, ranging from 81.6–97.2%.

**Effect of the noise level.** Increasing the smoothing noise $\sigma$ consistently trades a negligible drop at small radii for substantial gains at large radii across *every* baseline, yielding a tunable accuracy–robustness frontier. For example, on `CIFAR-10`, raising $\sigma$ from 0.25 to 0.50 leaves performance at $r = 0.75$ unchanged (44.3%) yet improves $r = 2.0$ from 8.52% to 12.5%. The same adjustment on `ImageNet` elevates $r = 2.0$ from 5.42% to 24.8%. These monotonic improvements confirm that `HyCAS` provides a controllable, rather than fixed, trade-off curve.

## 4.3 Empirical Adversarial Robustness

Across our empirical evaluations (Tables 3–4), `HyCAS` achieves the highest robust top-1 accuracy under APGD-20 and AA-20 at $\epsilon \in \{8, 16\}/255$. Specifically, on the `NIH-CXR` benchmark, `HyCAS` retains robust accuracy, outperforming the leading baseline (`CTRW`) by about **+1.8–4.2%** across these attacks while maintaining similar clean-set accuracy (89.5% vs. 88.4%). A similar trend appears on the `NCT-CRC-HE-100K` dataset, where `HyCAS` records robust accuracies of **76.7–79.3%** at $\epsilon = 16/255$ against the same attacks, exceeding `CTRW` by roughly **+1.5–2.6%** and leaving earlier certified defences (e.g., `ARS, DRS`) more than **+12%** behind at this stronger perturbation level. Dermoscopic `HAM10000` and fundus-image `EyePACS` exhibit the same hierarchy: `HyCAS` secures robust accuracies of **53.1–67.8%** against `APGD-20` and `AA-20` attacks on HAM10000—around **+1.9–3.1%** better than the next-best adversarial defence—and widens the margin on `EyePACS` to

**58.3–72.6%**, thereby surpassing the leading baseline `CTRW` by approximately **+2.1–2.8%**. *Together, these results show that `HyCAS` transfers its randomized Lipschitz strategy from certification to empirical regimes, maintaining clean accuracy while achieving state-of-the-art adversarial robustness.*

Under stronger `APGD` attacks on `CIFAR-10/100` (Figs. 2–3), `HyCAS` outperforms all baselines and preserves its advantage as attack strength increases. On `CIFAR-10`, when the perturbation radius is varied from $\epsilon = 0.01$ **to** $0.08$, `HyCAS` traces the upper envelope of the robust-accuracy curves, retaining an $\approx 10\%$ advantage at the largest perturbation, where leading methods collapse. A similar trend holds on `CIFAR-100` as the number of `APGD` iterations increases from **10 to 100**: all prior defenses, including `TRADES` (Zhang et al., 2019) and `HR` (Bennouna et al., 2023), degrade monotonically, whereas `HyCAS` declines more gracefully and remains **7–12%** above the closest competitor at 100 steps, confirming that its internally resampled attention noise and random projections thwart extended optimization. Thus, this randomized, Lipschitz-constrained design scales gracefully with both perturbation size and steps, offering adversarial robustness and a broader safety margin.

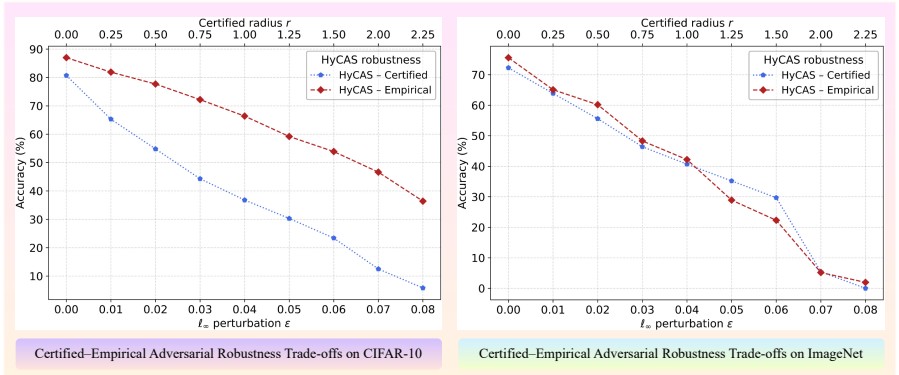

Figure 4: Trade-off between certified and empirical adversarial robustness achieved by `HyCAS` on the `CIFAR-10` (Left) and `ImageNet` (Right) datasets.

### 4.4 CERTIFIED–EMPIRICAL ROBUSTNESS TRADE–OFF

Figure 4 plots `HyCAS` on a three-axis Pareto frontier that couples certified $\ell_2$ accuracy (radius $r$) with empirical $\ell_\infty$ robustness (`APGD-20` accuracy at perturbation strength $\epsilon$). Across both `CIFAR-10` and ImageNet, the frontier is smooth and strictly downward-sloping: as the certified radius widens, empirical robustness inevitably contracts. Two consistent phenomena stand out: (a) **Certificate conservativeness.** For the small perturbation regime (left-most region), the empirical curve lies markedly above the certified curve, confirming that formal certificates are—by design—pessimistic relative to observed robustness. (b) **Norm mismatch tail-gap.** At large radii/perturbation strength (right-most region), the gap widens further, highlighting the inherent difficulty of translating $\ell_2$ guarantees into $\ell_\infty$ performance.

`HyCAS` achieves this trade-off by increasing the smoothing noise from $\sigma = 0.25$ to $0.50$ (arrow along each curve) leaves mid-radius performance virtually unchanged, yet extends both certified and empirical robustness deep into the high-perturbation regime. On `CIFAR-10`, certified accuracy at radius $r = 2.0$ improves from 8.5% to 12.5%, while ImageNet shows an even larger jump—from 5.4% to 24.8%—at the same radius. Crucially, these gains incur **minimal loss** in clean-accuracy / small-$\epsilon$ robustness, giving this *state-of-the-art adversarial defense* a **knob** to dial the desired security level without wholesale accuracy sacrifice. *See Appendix A.9 for additional experiments and Appendices A.10-A.11 for detailed ablations and certified and empirical robustness discussions.*

## 5 CONCLUSION

We presented HyCAS, a randomized adversarial defence whose deterministic 1-Lipschitz architecture is incorporated with two forms of data-independent internal randomness, yielding a global $\leq$ 2-Lipschitz network and a simple margin-based $\ell_2$ certificate. Experiments on diverse imaging benchmarks demonstrate state-of-the-art certified accuracy and strong empirical robustness against powerful $\ell_\infty$ attacks. Future work includes deriving tighter $\ell_\infty$ certificates, designing lighter-weight certification samplers, and integrating `HyCAS` into multi-modal clinical pipelines.

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

# A APPENDIX

## A.1 ADDITIONAL RELATED STUDY

Note that: in the *Table 5*, we *compare the properties for novel adversarial defense approach for enhancing adversarial robustness against existing baselines, demonstrating how* `HyCAS` *uniquely overcomes each identified research gap.*

## A.2 PRELIMINARIES

### A.2.1 RANDOMIZED SMOOTHING (RS)

Consider a $k$-class classification problem with input $x \in \mathbb{R}^d$ and label $y \in \mathcal{Y} = \{c_1, \ldots, c_k\}$. RS first corrupts each input $x$ by adding isotropic Gaussian noise $\mathcal{N}(\varepsilon; 0, \sigma^2 I)$. It then turns an arbitrary base classifier $f$ into a smoothed version $F$ that possesses $\ell_2$ certified robustness guarantees. The smoothed classifier $F$ returns whichever class the base classifier $f$ is most likely to return under the distribution $\mathcal{N}(x + \varepsilon; x, \sigma^2 I)$,

$$F(x) = \arg \max_{c \in \mathcal{Y}} \Pr\big(f(x + \varepsilon) = c\big). \tag{11}$$

**Theorem 2** (Cohen et al., 2019). *Let $f : \mathbb{R}^d \to \mathcal{Y}$ be any deterministic or random function, and let $F$ be the smoothed version defined in Equation equation 11. Let $c_A$ and $c_B$ be the most probable and runner-up classes returned by $F$ with smoothed probabilities $p_A$ and $p_B$, respectively. Then $F(x + \delta) = c_A$ for all adversarial perturbations $\delta$ satisfying*

$$\|\delta\|_2 \leq R', \quad R' = \tfrac{1}{2} \sigma \big(\Phi^{-1}(p_A) - \Phi^{-1}(p_B)\big),$$

*where $\Phi^{-1}$ is the inverse standard-Gaussian* CDF.

In Equation 2, $\Phi$ denotes the Gaussian cumulative distribution function (CDF) and $\Phi^{-1}$ signifies its inverse function. Theorem 1 indicates that the $\ell_2$ certified robustness provided by RS is closely linked to the base classifier's performance on the Gaussian distribution; a more consistent prediction within a given Gaussian distribution will return a stronger certified robustness. (The proof of Theorem 1 can be found in Appendix A.1.) It is not clear how to calculate $p_A$ and $p_B$ exactly when $f$ is a deep neural network, so Monte Carlo sampling is used to estimate the smoothed probability. The theorem also establishes that, when we assign $p_A$ a lower-bound estimate $\underline{p}_A$ and assign $p_B$ an upper-bound estimate with $\underline{p}_B = 1 - \underline{p}_A$, the radius $R'$ equals

$$R' = \sigma \Phi^{-1}(\underline{p}_A).$$

Equation (3) follows from $-\Phi^{-1}(1 - \underline{p}_A) = \Phi^{-1}(\underline{p}_A)$. The smoothed classifier $F$ is therefore guaranteed to return the constant prediction $c_A$ around $x$ within the $\ell_2$ ball of radius $R'$.

### A.2.2 SPECTRAL NORMALISATION OF CONVOLUTIONS

For a kernel $K \in \mathbb{R}^{k_h \times k_w \times C_{\text{in}} \times C_{\text{out}}}$ we denote by $\mathcal{C}_K$ the induced circular convolution. We follow the two most widely–used operator-norm estimators:

**(a) Exact Fourier bound** (Sedghi et al., 2018) derived

$$\sigma_\star(K) = \max_{\omega \in \Omega} \|\widehat{K}(\omega)\|_2, \qquad \|\mathcal{C}_K\|_{\text{op}} = \sigma_\star(K), \tag{12}$$

which we adopt verbatim in Eq. 12 to scale kernels whenever an FFT is affordable.

**(b) Power-iteration (PI) surrogate** (Miyato et al., 2018) proposed a light $T$-step estimate, also used by subsequent Lipschitz CNNs. Our implementation in Eq. 13 mirrors their update:

$$\hat{\sigma}^{(T)}(K) = \langle u^{(T)}, \mathcal{C}_K(v^{(T)})\rangle, \tag{13}$$

with $T = 5$ as in their default setting.

**Kernel rescaling.** Both estimators feed the same renormalisation rule

$$\widetilde{K} = \frac{K}{\max\{\hat{\sigma}(K), 1\} + \varepsilon}, \qquad \varepsilon = 10^{-6}, \tag{14}$$

which keeps $\|\mathcal{C}_{\widetilde{K}}\|_{\mathrm{op}} \leq 1$. (The clamp $\max\{\hat{\sigma}, 1\}$ is a minor safety tweak; we note it here for completeness but do not claim novelty.)

**Proposition 3** (Layer-wise 1-Lipschitzness). *Eqs. 12–14 ensure* $\|\mathcal{C}_{\widetilde{K}}\|_{\mathrm{op}} \leq 1$.

All subsequent sections treat Eq. 14 as a *black-box deterministic contraction*. Our contribution begins only after this step, in the following Method section.

*Proof.* Fix $\omega$ and let $x, y$ be arbitrary inputs. Define

$$z_x = \mathcal{C}_{\widetilde{K}}(x), \qquad z_y = \mathcal{C}_{\widetilde{K}}(y).$$

**Step 1: 1-Lipschitz contraction.** By construction of $\widetilde{K}$ (Eq. 14) we have

$$\|z_x - z_y\|_2 = \|\mathcal{C}_{\widetilde{K}}(x) - \mathcal{C}_{\widetilde{K}}(y)\|_2 \leq \|x - y\|_2. \tag{15}$$

**Step 2: Bounded multiplicative mask.** The mask generated by RANI is *data-independent* for fixed $\omega$, and satisfies the element-wise bound $M(\omega) \in [0, 1]^{H \times W \times C}$. Consequently

$$1 \leq 1 + M(\omega) \leq 2 \quad \text{(element-wise)}.$$

For any tensor $a$ this implies

$$\|(1 + M(\omega)) \odot a\|_2 \leq 2\|a\|_2. \tag{16}$$

**Step 3: Lipschitz constant of $F(\cdot, \omega)$.** Using definition equation 6,

$$F(x, \omega) - F(y, \omega) = (1 + M(\omega)) \odot (z_x - z_y),$$

and therefore, by equation 16 and equation 15,

$$\|F(x, \omega) - F(y, \omega)\|_2 \leq 2\|z_x - z_y\|_2 \leq 2\|x - y\|_2.$$

Because the bound holds for every choice of $x, y$, the mapping $F(\cdot, \omega)$ is 2-Lipschitz. $\qquad \square$

### A.2.3 RANDOM-PROJECTION FILTERS

Random-projection filters (RPF) replace a subset of convolution kernels with i.i.d. Gaussian weights. Let $x \in \mathbb{R}^{H \times W \times C_{\mathrm{in}}}$ be an input, $F \in \mathbb{R}^{k^2 C_{\mathrm{in}} \times C_{\mathrm{out}}}$ the flattened kernel matrix and $z = F^{\top} x$ the projected feature. When the number of random columns $C_{\mathrm{out}} = N_r$ satisfies the Johnson–Lindenstrauss lower bound,

$$(1 - \varepsilon)\|x_i - x_j\|_2^2 \leq \|z_i - z_j\|_2^2 \leq (1 + \varepsilon)\|x_i - x_j\|_2^2, \tag{17}$$

local geometry is provably preserved (Dong & Xu, 2023). A standard way to keep the mapping 1-Lipschitz is to rescale the frozen kernel with a spectral-norm estimate obtained by a few power-iteration (PI) steps after each forward pass.

## A.3 PROOFS

*Proof of Theorem 1.* By Propositions 2 and 5, SNCAN and RPFAN are $\leq$ 2-Lipschitz. By Proposition 1, FDPAN's gated output is also $\leq$ 2-Lipschitz. Finally, Proposition 4 shows the per-channel convex fusion has $\mathrm{Lip}(z) \leq \max_{b \in B} \mathrm{Lip}(G_b) \leq 2$. $\qquad \square$

**Proposition 4** (Convex fusion retains the max-Lipschitz). *Given channel-wise convex fusion $z(\cdot)$ (see Eq. 2) that satisfies $Lipschitz(z) \leq \max_b Lipschitz(G_b) \leq 2$, if every stream output is $\leq$ 2-Lipschitz, then the HyCAS block is also $\leq$ 2-Lipschitz.*

*Proof of Proposition 4.* Fix a channel $c$ and any $x, y$. Triangle inequality gives

$$\|z(x)_{:,:,c} - z(y)_{:,:,c}\| = \left\| \sum_b \alpha_{b,c}(G_b(x)_{:,:,c} - G_b(y)_{:,:,c}) \right\| \leq \sum_b \alpha_{b,c} \|G_b(x) - G_b(y)\|.$$

Since $\|G_b(x) - G_b(y)\| \leq L_b\|x - y\|$ and $\sum_b \alpha_{b,c} = 1$, we have $\|z(x)_{:,:,c} - z(y)_{:,:,c}\| \leq (\max_b L_b)\|x - y\|$. Taking the maximum over channels yields $\text{Lip}(z) \leq \max_b L_b$, Thus $Lip(z) \leq 2$. $\qquad\square$

**Lemma 2** (Expectation preserves Lipschitz constant). *If $x \mapsto s(x, \Omega)$ is L-Lipschitz for all $\Omega$, then the expected logits $Z(x) = \mathbb{E}_\omega[s(x, \omega)]$ are L-Lipschitz. (By Jensen's inequality and linearity of expectation (Dong & Xu, 2023).) Hence,* `HyCAS`'*s expected classifier inherits the same constant.*

*Proof of Lemma 2.* For any $x, y$,

$$\|Z(x) - Z(y)\| = \|\mathbb{E}_\omega[s(x, \omega) - s(y, \omega)]\| \leq \mathbb{E}_\omega \|s(x, \omega) - s(y, \omega)\| \leq \mathbb{E}_\omega[L\|x - y\|] = L\|x - y\|,$$

using Jensen's inequality $\|\mathbb{E}X\| \leq \mathbb{E}\|X\|$. $\qquad\square$

*Proof of Corollary 1.* By Lemma 2, $Z$ is 2-Lipschitz. Hence for any coordinate $c$,

$$|Z_c(x + \delta) - Z_c(x)| \leq \|Z(x + \delta) - Z(x)\|_\infty \leq \|Z(x + \delta) - Z(x)\|_2 \leq 2\|\delta\|_2 < \frac{\Delta(x)}{2}.$$

Thus the top logit can decrease by at most $\Delta/2$ and the runner-up can increase by at most $\Delta/2$; their order cannot swap. $\qquad\square$

*Proof of Proposition 1.* By the triangle inequality and the chain rule for Lipschitz maps,

$$\begin{aligned}
\|G_{\text{FDPAN}}(x; \omega) - G_{\text{FDPAN}}(y; \omega)\| &= \|H(x) - H(y) + R(H(x); \omega) - R(H(y); \omega)\| \\
&\leq \|H(x) - H(y)\| + \|R(H(x); \omega) - R(H(y); \omega)\| \\
&\leq \text{Lip}(H)\|x - y\| + \text{Lip}(R)\|H(x) - H(y)\| \\
&\leq 1 \cdot \|x - y\| + 1 \cdot 1 \cdot \|x - y\| = 2\|x - y\|.
\end{aligned}$$

Thus $\text{Lip}(G_{\text{FDPAN}}) \leq 2$. $\qquad\square$

*Proof of Proposition 2.* Let $x, y \in \mathbb{R}^{H \times W \times C}$ and set $z_1 = C_{K_e}(x)$ and $z_2 = C_{K_e}(y)$. Using equation 5 and the sub-multiplicativity of operator norms,

$$\begin{aligned}
\|G_{\text{SNCAN}}(x; M_\omega) - G_{\text{SNCAN}}(y; M_\omega)\|_2 &= \|(I + D_\omega)(z_1 - z_2)\|_2 \\
&\leq \|I + D_\omega\|_2 \|z_1 - z_2\|_2 \\
&\leq 2\|C_{K_e}(x) - C_{K_e}(y)\|_2 \\
&\leq 2\|x - y\|_2,
\end{aligned}$$

which establishes the claim. $\qquad\square$

**Proposition 5** (`RPFAN` is 2-Lipschitz). *Let $U$ be an orthogonal $1 \times 1$ channel mixer ($\|U\|_{\text{op}} = 1$). Let $W_{SN}$ be a spectrally normalized random-projection filter so that the linear map $H_{\text{RPFAN}}(x) := \text{Conv}(Ux; W_{SN})$ satisfies $\text{Lip}(H_{\text{RPFAN}}) \leq 1$. Let $D_\omega = \text{diag}(M_\omega)$ with $M_\omega \in [0,1]^d$ and define*

$$G_{\text{RPFAN}}(x; M_\omega) = H_{\text{RPFAN}}(x) + D_\omega H_{\text{RPFAN}}(x) = (I + D_\omega)H_{\text{RPFAN}}(x).$$

*Then $\text{Lip}(G_{\text{RPFAN}}) \leq 2$.*

*Proof of Proposition 5.* $\|G(x) - G(y)\| = \|(I + D_\omega)(H_{\text{RPFAN}}(x) - H_{\text{RPFAN}}(y))\| \leq \|I + D_\omega\|_2 \text{Lip}(H_{\text{RPFAN}})\|x - y\| \leq 2 \cdot 1 \cdot \|x - y\|$. $\qquad\square$

*Proof of Lemma 1.* Fix any realisation of the noise $\omega$ and set $D_\omega = \text{diag}(M_\omega) \in \mathbb{R}^{d \times d}$. By Eq. equation 9 the `RANI` transformation satisfies

$$R(h; M_\omega) = h + D_\omega h = (I + D_\omega) h.$$

**Step 1: bound the operator norm of $I + D_\omega$.** Because every coordinate of $M_\omega$ lies in $[0, 1]$, each diagonal entry of $D_\omega$ is in the same interval. Hence all singular values of $D_\omega$ are $\leq 1$ and

$$\|I + D_\omega\|_2 \leq \|I\|_2 + \|D_\omega\|_2 = 1 + 1 = 2.$$

**Step 2: translate the norm bound into a Lipschitz constant.** For arbitrary $h_1, h_2 \in \mathbb{R}^d$,

$$\begin{aligned}
\|R(h_1; M_\omega) - R(h_2; M_\omega)\|_2 &= \|(I + D_\omega)(h_1 - h_2)\|_2 \\
&\leq \|I + D_\omega\|_2 \|h_1 - h_2\|_2 \\
&\leq 2 \|h_1 - h_2\|_2.
\end{aligned}$$

Therefore $R(\cdot; M_\omega)$ is 2-Lipschitz with respect to the Euclidean norm for every draw of $\omega$, completing the proof. $\square$

**Lemma 3** (Orthogonal transforms are 1-Lipschitz). *If $U \in \mathbb{R}^{d \times d}$ is orthonormal then $Lip(U) = 1$. In particular, 2-D `DCT`/`IDCT` and any frozen orthogonal $1 \times 1$ convolution satisfy $Lip = 1$.*

For a map $h : \mathbb{R}^d \to \mathbb{R}^d$ the $\ell_2$–Lipschitz constant is

$$\text{Lip}(h) = \sup_{u \neq v} \frac{\|h(u) - h(v)\|_2}{\|u - v\|_2}.$$

Throughout we use the vectorised $\ell_2$ norm over $N \times H \times W \times C$ tensors. We make repeated use of: *Triangle inequality.* $\|a + b\|_2 \leq \|a\|_2 + \|b\|_2$. *Convex combination bound.* If $\sum_i \lambda_i = 1$ and $\lambda_i \geq 0$ then $\left\|\sum_i \lambda_i a_i\right\|_2 \leq \sum_i \lambda_i \|a_i\|_2$. *Jensen.* $\|\mathbb{E}[X]\|_2 \leq \mathbb{E}[\|X\|_2]$.

*Proof of Lemma 3.*

$$\|Ux - Uy\| = \|U(x - y)\| = \|x - y\| \qquad \text{for all } x, y.$$

$\square$

**Lemma 4** (Spectral normalisation). *Rescaling a convolutional kernel $W$ by $W/\max(\|W\|_2, 1)$ enforces $\text{Lip}(\text{Conv}_W) \leq 1$ (Gouk et al., 2021).*

*Proof of Lemma 4.* By construction,

$$\|C_{K_e}\|_{\text{op}} = \frac{\|C_K\|_{\text{op}}}{\max\{\sigma^{(K)}, 1\}} \leq \frac{\max\{\|C_K\|_{\text{op}}, \sigma^{(K)}\}}{\max\{\sigma^{(K)}, 1\}} \leq 1.$$

$\square$

### A.4 CERTIFIED PREDICTION UNDER HYCAS

Each `HyCAS` stream comprises a 1-Lipschitz deterministic core followed by a *data-independent* `RANI` module. Conditioning on the internal noise $\omega$, each stream is therefore 2-Lipschitz (see Lemma 3), and the composite core remains 2-Lipschitz (ref. Lemma 1). Specifically, the `FDPAN` stream is the only exception: it contains two residual blocks (`SNCAN` + `RANI`), giving a naïve 4—Lipschitz upper bound. We tighten this to $\leq 2$-Lipschitz by scaling the skip connection (Proposition 1). A convex channel gate then fuses the streams without increasing the Lipschitz constant (Proposition 4). Finally, stacking modules and applying a global calibrator with gain $c \leq 2/\widehat{L}_{\text{net}}$ ensures the entire network is at most 2—Lipschitz.

Table 5: Scope of representative certified, empirical, and hybrid defences. A ✓ indicates that the property is explicitly addressed, or the domain is reported, in the original paper.

| Method | Certified | Empirical | Natural images | Medical images |
|---|---|---|---|---|
| RS (Cohen et al., 2019) | ✓ | | ✓ | |
| IRS (Ugare et al.) | ✓ | | ✓ | |
| DRS (Xia et al., 2024) | ✓ | | ✓ | |
| ARS (Lyu et al., 2024) | ✓ | | ✓ | |
| LOT (Xu et al., 2022) | ✓ | | ✓ | |
| SLL (Araujo et al., 2023) | ✓ | | ✓ | |
| PNI (He et al., 2019) | | ✓ | ✓ | |
| Learn2Perturb (Jeddi et al., 2020) | | ✓ | ✓ | |
| CTRW (Ma et al., 2023) | | ✓ | ✓ | |
| RPF (Dong & Xu, 2023) | | ✓ | ✓ | |
| CAP (Xiang et al., 2023) | | ✓ | | ✓ |
| **HyCAS (ours)** | ✓ | ✓ | ✓ | ✓ |

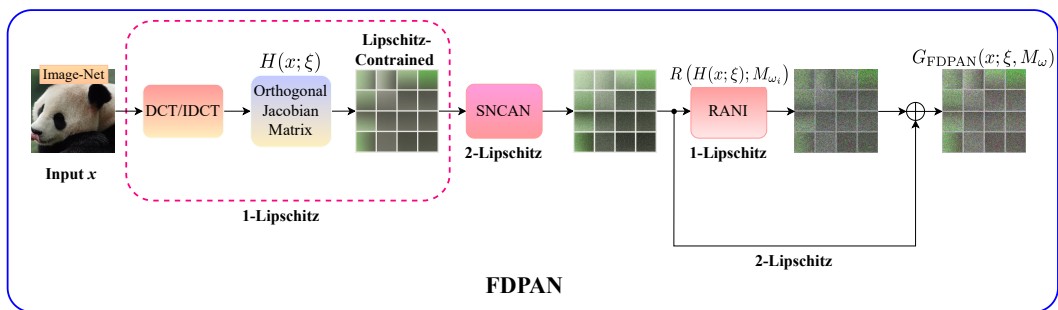

Figure 5: Overview of FDPAN stream. A four-stage cascade: (i) low-pass DCT masking and orthogonal 1×1 channel mix (both 1-Lipschitz); (ii) SNCAN block (spectrally normalized convolution) with RANI; (iii) additional RANI; and (iv) skip/gating. The stream remains $\leq$2-Lipschitz.

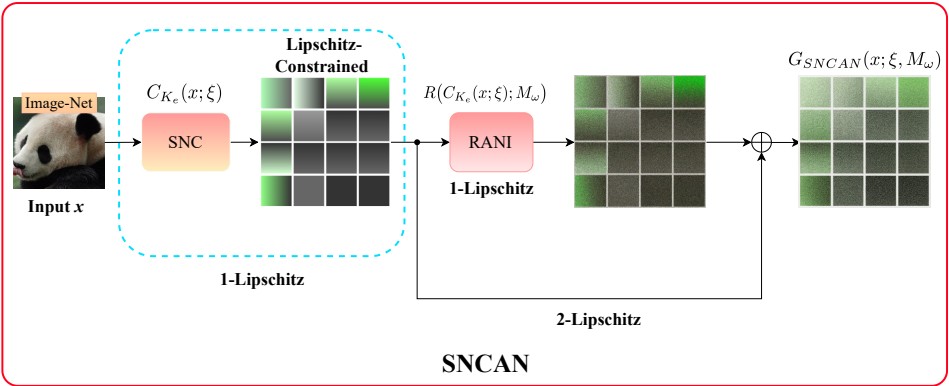

Figure 6: Overview of SNCAN block. A spectrally normalized convolution ($C_{K_e}$) ensures operator norm $\leq 1$; RANI applies a bounded, data-independent attention mask $M_\omega$ so the block output equals $(I + D_\omega) C_{K_e}(x)$, which is $\leq 2$-Lipschitz.

**Margin certificate.** Define the *expected logits*, averaged only over the model's internal randomness, be

$$Z(x) = \mathbb{E}_\omega[s(x; \omega)], \ \mathrm{Lip}(Z) \leq 2 \ \text{(Lemma 2)}.$$

Let $\Delta_Z(x) = Z_{(1)}(x) - Z_{(2)}(x)$ denote the gap between the top two logits. The certified $\ell_2$ radius at $x$ is then,

$$r_2(x) = \frac{\Delta_Z(x)}{4},$$

which guarantees $\arg\max Z(x + \delta) = \arg\max Z(x)$ for every perturbation $\|\delta\|_2 < r_2(x)$ (Corollary 1). For $\ell_\infty$, the norm inequality $\|\delta\|_2 \leq \sqrt{d}\|\delta\|_\infty$ yields the conservative certificate $r_\infty(x) = \frac{r_2(x)}{\sqrt{d}}$.

**Estimation under internal randomness.** At test time we approximate $Z$ through Monte Carlo sampling over $\omega$. Draw $n_0$ pilot samples to identify the top class, then take $n$ additional samples; compute one-sided confidence bounds for $Z_{(1)}(x)$ and $Z_{(2)}(x)$ and certify with

$$r_{\text{LCB}}(x) = \frac{\text{LCB}\big(Z_{(1)}(x)\big) - \text{UCB}\big(Z_{(2)}(x)\big)}{4},$$

at confidence $1 - \alpha$.

## A.5 EXTENDED DETAILS FOR FDPAN, SNCAN, AND RPFAN STREAMS

Detailed overviews of the three parallel streams—FDPAN, SNCAN, and RPFAN—are illustrated in Figs. 5–7, respectively.

## A.6 EXTENDED DETAILS FOR RANI MODULE

**Deterministic attentions.** We design two deterministic attentions—*local* (LA) and *channel* (CA)—that highlight informative local and inter-channel dependencies, respectively. Specifically, we leverage GAP and 1*1 convolution followed by leveraging dense layer and sigmoid to learn these attention maps:

$$LA(z) = \sigma\big(\text{Conv}_{1\times 1}^{(l)}(z)\big),\ CA(z) = \sigma\Big(\text{Dense}_2^{(l)}\big(\text{ReLU}\big(\text{Dense}_1^{(l)}(\text{GAP}(z))\big)\big)\Big), \tag{18}$$

**Injecting stochasticity.** We these deterministic attention maps into randomized attention maps ($\gamma_g'$ and $\gamma_l'$) via injecting feature layer noises $\eta_{I_g}, \eta_{I_l} \in \mathbb{R}^C$, thereby incorporating stochasticity. We formulate this as: $\gamma_g' = \eta_{I_g} + g',\quad \gamma_l' = \eta_{I_l} + l',$ (19) where $\sigma_g, \sigma_l \in \mathbb{R}^C$ are trainable scale vectors and $\Psi(u) = \min\{1, \max\{0, u\}\}$ clips the maps into $[0,1]$.

**Noise parameterization and iterative refinement.** To realise heteroscedastic, yet data-independent, stochasticity at minimal cost we employ a two-step self-modulation loop

$$\eta_{I_\bullet} = \eta_\bullet \odot \big(\sigma_\bullet + \eta_\bullet \odot \sigma_\bullet\big), \qquad \bullet \in \{g, l\},$$

This yields two potential benefits: (a) **Richer expressivity**—because $\sigma_\bullet$ is trainable, the model learns which channels benefit from strong noise and which should stay nearly deterministic; and (b) **Negligible overhead**—only $2C$ extra scalars per branch.

**Iterative noise fusion.** We propagate the stochastic smoothing through the backbone in *four* stages. At each stage $j \in \{1, \ldots, 4\}$ the current feature tensor $x_f'$ is modulated by the noisy randomized the deterministic attention maps, $\Psi(\gamma_g')_j$ and $\Psi(\gamma_l')_j$ followed by fuse them as

$$x_U = x_f' \odot \prod_{j=1}^{4} \big[\Psi(\gamma_g')_j \odot \Psi(\gamma_l')_j\big], \tag{20}$$

Each stage injects a freshly resampled attention noise yielding a progressively smoother–stochastic feature tensor. This cascade progressively smooths the feature tensor and presents a continually shifting optimisation landscape to an adversary, thereby enhancing robustness while preserving the global 2-Lipschitz guarantee. Thus RANI converts every deterministic 1-Lipschitz block into a randomised counterpart that keeps the certified $\ell_2$ margin while impeding adversarial attacks by presenting a moving optimisation landscape.

## A.7 DETAILS ABOUT THE HyCAS CERTIFIED ALGORITHM

This section illustrates the details of the certified algorithm of `HyCAS`, as shown in the following subsections:

### A.7.1 HyCAS TRAINING

Our base network couples a *deterministic, Lipschitz–constrained backbone* with *stochastic smoothing branches*. Concretely, let $f_\theta(\cdot; \Omega)$ denote the hybrid classifier with parameters $\theta$ and internal randomness $\Omega = (\psi, M_\omega)$, where $\psi$ parametrizes implicit randomness (e.g., random projection filters) and $M_\omega$ injects explicit attention noise. Following randomized smoothing (RS), we expose the input to isotropic Gaussian noise $\varepsilon \sim \mathcal{N}(0, \sigma^2 I)$ during training and minimize the expected loss

$$\min_\theta \ \mathbb{E}_{(x,y)} \mathbb{E}_{\varepsilon \sim \mathcal{N}(0,\sigma^2 I), \, \Omega} \ \ell(f_\theta(x + \varepsilon; \Omega), \, y),$$

which is the same objective used to train the stochastic component in our hybrid architecture (cf. Eq. 10 for `RANI` in `HyCAS`). To make RS effective at scale while retaining deterministic control, the backbone is constrained to be $L_{\text{net}} \le 2$-Lipschitz; our implementation mirrors the `HyCAS` construction where residual blocks are scaled so the stacked network remains $\le 2$-Lipschitz and thus amenable to margin certification.

To mitigate the curse of dimensionality inherent to RS, we optionally activate a DRS (Dual Randomized Smoothing) path that partitions $x$ into two lower-dimensional sub-inputs and smooths them separately before fusion. This preserves most information while tightening the $\ell_2$ certificate upper bound from $O(1/\sqrt{d})$ to $O(1/\sqrt{m} + 1/\sqrt{n})$ with $m + n = d$. Our training simply shares the same $\theta$ and minimizes the same expectation, with the forward pass executing the two DRS branches in parallel.

Algorithm 1 summarizes one epoch: for each minibatch we (i) sample $(\varepsilon, \Omega)$ once per forward; (ii) run the deterministic Lipschitz backbone and the stochastic streams; (iii) backpropagate the Monte-Carlo estimate of the RS objective; and (iv) apply the Lipschitz constraints (spectral normalization / calibrated residual scaling) to keep $L_{\text{net}} \le 2$.

---

**Algorithm 1** `HyCAS` Training

**Requires:** Dataset $\mathcal{D}$; epochs $E$; batch size $B$; noise level $\sigma$; HyCAS-integrated network $f_\theta$ with streams $\{\text{SNCAN}, \text{RPFAN}, \text{FDPAN}\}$, convex channel gate $\alpha_{b,c}$ $(\sum_b \alpha_{b,c} = 1)$, and 1-Lipschitz building blocks; optimizer $\mathcal{O}$; (optional) stream loss weights $\zeta, \phi, \nu, \kappa$.

1 **Init:** Initialize $\theta$; set spectral normalisation (SN) for all convs (operator norm $\le 1$); **for** $e = 1$ **to** $E$ **do**

2      **foreach** *minibatch* $\{(x_i, y_i)\}_{i=1}^B \sim \mathcal{D}$ **do**
         `// Resample internal randomness once per minibatch (HyCAS execution protocol)`

3          Resample random-projection filters for RPFAN and attention-noise masks for all streams, collect as $\Omega$. `// RS-style training noise at the input`

4          **for** $i = 1$ **to** $B$ **do**

5              Draw $\varepsilon_i \sim \mathcal{N}(0, \sigma^2 I)$; set $\tilde{x}_i \leftarrow x_i + \varepsilon_i$.
         `// Forward through the three streams + convex fusion (each stream` $\le 2$`-Lipschitz)`

6          Compute per-stream feature maps $G_b(\tilde{x}_i; \Omega)$ for $b \in \{\text{SNCAN}, \text{RPFAN}, \text{FDPAN}\}$. Fuse per channel: $z(\tilde{x}_i)\, \texttt{:}\, c \leftarrow \sum_b \alpha_{b,c} G_b(\tilde{x}_i; \Omega)\, \texttt{:}\, c$. `// Loss: single fused CE, or the HyCAS-weighted multi-branch objective`

7          $L \leftarrow \kappa \mathcal{L}(z(\tilde{x}_i), y_i) + \zeta \mathcal{L}(G_{\text{FDPAN}}, y_i) + \phi \mathcal{L}(G_{\text{SNCAN}}, y_i) + \nu \mathcal{L}(G_{\text{RPFAN}}, y_i)$. Update $\theta \leftarrow \mathcal{O}\big(\theta, \nabla_\theta \frac{1}{B} \sum_i L\big)$. `// Keep layer-wise operator norms` $\le 1$ `(SN) to maintain global` $\le 2$`-Lipschitz envelope`

8          Re-apply SN to all conv kernels.

     `// Final global calibrator (gain) to cap the network Lipschitz constant by 2`

9 Estimate $L_{\text{net}}$ (product of per-block bounds); scale last linear by $\gamma \leftarrow \min(1, 2/L_{\text{net}})$.

---

---

**Algorithm 2 HyCAS Inference and Certification**

---

**Input:** Trained `HyCAS`-integrated classifier $f_\theta$ (globally $\leq 2$-Lipschitz); test point $x$; class set $\mathcal{Y}$ of size $K$; Gaussian noise level $\sigma$; pilot $n_0$ and budget $n$ for RS; significance $\alpha$ (set $\alpha_{\mathrm{RS}} = \alpha_{\mathrm{Lip}} = \alpha/2$).

**Output:** Certified label $\hat{y}$ and radius $R > 0$, or ABSTAIN.

10 **(A) RS branch (standard randomized smoothing).**     `// Cohen-style certificate; pilot then CI`

11 **for** $i = 1$ **to** $n_0$ **do**

12    Draw $\varepsilon \sim \mathcal{N}(0, \sigma^2 I)$ and internal randomness $\Omega$;   $c_i \leftarrow f_\theta(x + \varepsilon; \Omega)$;

13 Let $\hat{c}_A \leftarrow \arg\max_{c \in \mathcal{Y}} \mathrm{count}_{n_0}(c)$. **for** $i = 1$ **to** $n$ **do**

14    Draw $\varepsilon \sim \mathcal{N}(0, \sigma^2 I)$ and $\Omega$;   $c_i \leftarrow f_\theta(x + \varepsilon; \Omega)$;

15 Let $m \leftarrow \mathrm{count}_n(\hat{c}_A)$ and $\hat{p}_A^{\mathrm{LB}} \leftarrow \text{CLOPPERPEARSONLOWER}(m, n, 1 - \alpha_{\mathrm{RS}})$. **if** $\hat{p}_A^{LB} \leq \frac{1}{2}$ **then** set $R_{\mathrm{RS}} \leftarrow 0$

16 **else** $R_{\mathrm{RS}} \leftarrow \sigma\, \Phi^{-1}\big(\hat{p}_A^{\mathrm{LB}}\big)$;   $\hat{y}_{\mathrm{RS}} \leftarrow \hat{c}_A$.

                   `// `$\Phi^{-1}$` is the standard normal inverse CDF`

17 **(B) Lipschitz-margin branch (deterministic certificate).**      `// HyCAS margin certificate`

18 *Freeze* internal randomness $\Omega^\star$ (fix seeds), and compute logits $s(\cdot; \Omega^\star)$.   $\hat{y}_{\mathrm{Lip}} \leftarrow \arg\max_{c \in \mathcal{Y}} s_c(x; \Omega^\star)$;   $s^{(1)} \leftarrow \max_c s_c(x; \Omega^\star)$;   $s^{(2)} \leftarrow \max_{c \neq \hat{y}_{\mathrm{Lip}}} s_c(x; \Omega^\star)$. **if** $s^{(1)} \leq s^{(2)}$ **then** set $R_{\mathrm{Lip}} \leftarrow 0$

19 **else** $R_{\mathrm{Lip}} \leftarrow \dfrac{s^{(1)} - s^{(2)}}{4}$

                 `// Since `$\mathrm{Lip}(f_\theta) \leq 2$`, radius is (margin)/(2·Lip)`

20 **(C) Pick the stronger valid certificate. if** $\max(R_{\mathrm{RS}}, R_{\mathrm{Lip}}) = 0$ **then return** ABSTAIN

21 **else if** $R_{\mathrm{RS}} \geq R_{\mathrm{Lip}}$ **then return** $(\hat{y}_{\mathrm{RS}}, R_{\mathrm{RS}})$

22 **else return** $(\hat{y}_{\mathrm{Lip}}, R_{\mathrm{Lip}})$

---

### A.7.2 HyCAS Inference-time Certification

At test time we provide two *independent* certificates, both for the exact network we evaluate:

1. **RS certificate.** We certify the smoothed classifier

$$g_\sigma(x) \triangleq \arg\max_{c \in \mathcal{Y}} \mathbb{P}_{\varepsilon, \Omega}[f_\theta(x + \varepsilon; \Omega) = c].$$

   We follow the standard two–stage Monte-Carlo protocol: draw $n_0$ samples to select the candidate class $\hat{c}$ and then $n$ samples to bound its probability. Let $\hat{p}_A$ and $\hat{p}_B$ be the empirical proportions of the top and runner-up classes. Using exact Clopper–Pearson intervals we obtain a $(1 - \alpha)$ lower bound $p_A^{\mathrm{LB}}$ on the top class and an upper bound $p_B^{\mathrm{UB}}$ on the second. If $p_A^{\mathrm{LB}} \leq \frac{1}{2}$ we abstain; otherwise the certified $\ell_2$ radius is

$$r_2^{\mathrm{RS}}(x) = \frac{\sigma}{2}\left(\Phi^{-1}(p_A^{\mathrm{LB}}) - \Phi^{-1}(p_B^{\mathrm{UB}})\right),$$

   where $\Phi^{-1}$ is the standard normal quantile. This is the tight Cohen–Rosenfeld–Kolter bound specialized and re-derived in `ARS` (via $f$-DP). In our experiments we mirror `DRS` sampling defaults ($n_0 = 100$, $n = 10^5$, $\alpha = 10^{-3}$). When the `DRS` path is enabled, class probabilities are estimated branch-wise and fused as in `DRS` before applying the same formula.

2. **Deterministic Lipschitz (margin) certificate.** Independently of input noise, we certify the *backbone + internal randomness* by averaging logits only over $\Omega$:

$$Z(x) \triangleq \mathbb{E}_\Omega[s(x; \Omega)], \qquad \text{with } \mathrm{Lip}(Z) \leq 2.$$

   Let $\Delta Z(x) = Z_{(1)}(x) - Z_{(2)}(x)$ be the gap between the top-two expected logits. Then for every perturbation $\|\delta\|_2 < \Delta Z(x)/4$, the $\arg\max$ of $Z(\cdot)$ is invariant; i.e., the model's prediction is certifiably robust within radius

$$r_2^{\mathrm{Lip}}(x) = \frac{\Delta Z(x)}{4}, \qquad r_\infty^{\mathrm{Lip}}(x) = \frac{r_2^{\mathrm{Lip}}(x)}{\sqrt{d}}.$$

We estimate $Z$ via Monte-Carlo over $\Omega$ (no input noise), exactly as recommended in `HyCAS`.

Algorithm 2 implements both procedures. In reporting, we return *two* radii $\left(r_2^{\text{RS}}(x),\ r_2^{\text{Lip}}(x)\right)$ for the same input $x$. Both are valid and interpretable: the first certifies the `RS/DRS-smoothed` classifier, the second certifies the *Lipschitz hybrid backbone averaged over internal noise*. This mirrors the practice in `ARS/RS` (majority-vote certificate) and `HyCAS` (margin certificate) while respecting their assumptions.

### A.8 EXTENDED EVALUATION SETUP

**Network Execution.** At the start of every mini-batch we resample, for each forward pass, (i) the attention-noise $M_\omega$ for {`FDPAN`, `SNCAN`, `RPFAN`} and (ii) the random projection filters for `RPFAN`. These samples stay fixed while adversarial examples are generated. At inference stage, for each test image, we draw one fresh set $(\psi, \omega)$, and evaluate `HyCAS` against adversarial attacks to ensure adversarial robustness.

**Implementation details.** Following (Cohen et al., 2019; Lyu et al., 2024; Xia et al., 2024), we use `ResNet-110` (He et al., 2016) on CIFAR-10/100 and `ResNet-50` on remaining imaging datasets (e.g., `ImageNet` (Deng et al., 2009), `CelebA` (Liu et al., 2015)), `NCT-CRC-HE-100K` (Kather et al., 2018) etc., as base classifiers for all training strategies. We report the best performance separately for a more comprehensive and fair comparison. We evaluate on `CIFAR-10/100`, `ImageNet-1k`, `CelebA` (unaligned, cropped attribute), `NCT-CRC-HE-100K`, `NIH CXR`, `EyePACS`, and `HAM10000`.

**For Certified Defense.** `HyCAS` certifies via a margin bound under an *at-most* 2-Lipschitz network. Let $Z(x) = \mathbb{E}_\omega[z(x)]$ denote the classifier averaged over the model's internal randomness; since $\text{Lip}(Z) \leq 2$, a pointwise certificate is $r(x) = \frac{\Delta_Z(x)}{4}$, $\Delta_Z(x) = Z_{(1)}(x) - Z_{(2)}(x)$. To estimate $Z(x)$ we Monte Carlo sample only the model's *internal* noise at inference. Unless noted otherwise, we take a pilot of $n_0 = 100$ samples to select the top class, then draw $n = 100{,}000$ additional samples to form one-sided confidence bounds and report $r_{\text{LCB}}(x) = \frac{\text{LCB}\left(Z_{(1)}(x)\right) - \text{UCB}\left(Z_{(2)}(x)\right)}{4}$ at confidence $1 - \alpha$ with $\alpha = 0.001$. During inference we draw a fresh $\omega$ on each forward pass. To control runtime for Monte Carlo estimation, we use a fixed rule per dataset:

- `CIFAR-10`: certify every $5^{\text{th}}$ test image (default settings $n_0$=100, $n$=100,000, $\alpha$=0.001).
- `ImageNet-1k`: certify every $100^{\text{th}}$ test image (default settings $n_0$=100, $n$=100,000, $\alpha$=0.001).
- `CelebA` (ARS-style): certify a *uniform, label-stratified* subset of 200 test images using $n_0$=100, $n$=50,000, and failure probability 0.05 (i.e., 95% confidence).
- `NCT-CRC-HE-100K`, `NIH ChestX-ray14`, `EyePACS`: certify a *uniform, label-stratified* subsample per dataset sized to yield $\approx$2,000 certified examples each (default settings $n_0$=100, $n$=100,000, $\alpha$=0.001; exact counts in the appendix).
- `HAM10`: certify the full test split when feasible; otherwise a uniform, label-stratified subsample (default settings $n_0$=100, $n$=100,000, $\alpha$=0.001; exact count in the appendix).

We sweep $\sigma \in \{0.25, 0.50, 1.0\}$ for comparability across settings.

**For Empirical Defense.** We follow the protocol of `SOTA` adversarial training strategy (Rice et al., 2020) to set up our experiments on our diverse datasets. For Adversarial Evaluation—`HyCAS` is tested under white-box attacks—*PGD (Madry et al., 2018), APGD (Croce & Hein, 2020), and* `AutoAttack` *(AA) (Croce & Hein, 2020)* using $\epsilon = \{\frac{8}{255}, \frac{16}{255}\}$, step size $\alpha = \frac{20}{255}$, and 10–100 iterations.

**Training Details for Certified Robustness.** Following `ARS`, we use a single recipe per dataset and train all `HyCAS`-integrated backbones. Inputs are perturbed *during training only* with i.i.d. Gaussian noise $\mathcal{N}(0, \sigma^2)$ (the same $\sigma$ as at certification). For `CIFAR-10`, we train for 200 epochs with a batch size of 256 using `AdamW` as the optimizer with learning rate $10^{-2}$ and weight decay $10^{-4}$. A step scheduler is used with step size 30 and decay factor $\gamma = 0.1$. For `CelebA, NCT-CRC-HE-100K, NIH CXR, EyePACS, and HAM10000`, we train for 200 epochs with a batch size of 64 using `SGD` as the optimizer with learning rate $5 \times 10^{-2}$. A step scheduler is used with step size 3 and decay factor $\gamma = 0.8$. For `ImageNet-1k`, we train for 200 epochs

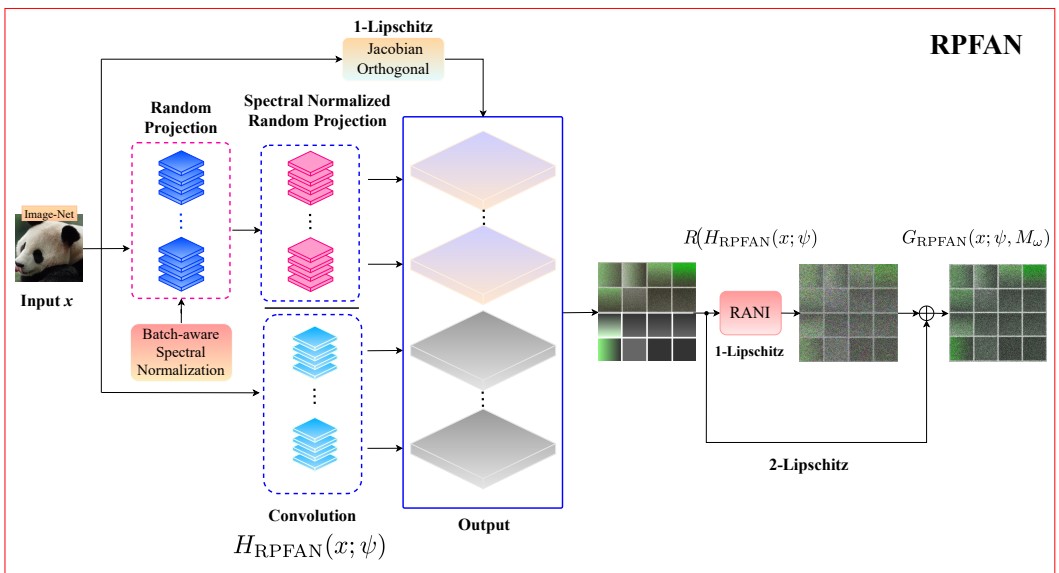

Figure 7: Overview of RPFAN stream. (i) Orthogonal 1×1 pre-mix (1-Lipschitz). (ii) Batch-aware spectral normalization of a random-projection convolution (1-Lipschitz core). (iii) RANI residual, yielding a $\leq$ 2-Lipschitz stochastic block.

Table 6: Robust accuracy (%) against $\ell_\infty$ attacks (PGD-20 and AA-20) on `NIH-CXR` (left) and `NCT-CRC-HE-100K` (right) at $\epsilon \in \{8/255, 16/255\}$.

| Method | NIH-CXR | | | | | NCT-CRC-HE-100K | | | | |
|---|---|---|---|---|---|---|---|---|---|---|
| | Clean | PGD–20 | | AA–20 | | Clean | PGD–20 | | AA–20 | |
| | | 8/255 | 16/255 | 8/255 | 16/255 | | 8/255 | 16/255 | 8/255 | 16/255 |
| AT | $89.1 \pm 1.91$ | $78.3 \pm 2.82$ | $68.4 \pm 3.62$ | $74.2 \pm 2.93$ | $64.1 \pm 3.70$ | $92.2 \pm 1.82$ | $80.4 \pm 2.72$ | $70.8 \pm 3.52$ | $76.3 \pm 2.83$ | $66.2 \pm 3.61$ |
| RPF | $88.4 \pm 1.82$ | $84.6 \pm 2.62$ | $73.2 \pm 3.42$ | $82.5 \pm 2.71$ | $70.8 \pm 3.52$ | $91.1 \pm 1.71$ | $87.5 \pm 2.51$ | $76.6 \pm 3.33$ | $84.2 \pm 2.62$ | $72.4 \pm 3.41$ |
| CTRW | $88.4 \pm 1.73$ | $85.7 \pm 2.41$ | $74.4 \pm 3.22$ | $84.5 \pm 2.48$ | $72.6 \pm 3.41$ | $90.4 \pm 1.62$ | $89.6 \pm 2.33$ | $77.5 \pm 3.12$ | $86.7 \pm 2.44$ | $75.2 \pm 3.22$ |
| DCS | $87.2 \pm 2.05$ | $83.5 \pm 2.65$ | $72.3 \pm 3.30$ | $81.7 \pm 2.72$ | $69.6 \pm 3.45$ | $90.3 \pm 1.93$ | $86.5 \pm 2.62$ | $75.2 \pm 3.35$ | $83.3 \pm 2.74$ | $71.6 \pm 3.46$ |
| ARS | $84.8 \pm 2.22$ | $77.2 \pm 2.95$ | $66.5 \pm 3.62$ | $72.8 \pm 3.11$ | $62.8 \pm 3.72$ | $86.8 \pm 2.14$ | $78.9 \pm 2.92$ | $68.3 \pm 3.61$ | $74.6 \pm 3.11$ | $64.5 \pm 3.73$ |
| DRS | $83.9 \pm 2.33$ | $76.2 \pm 2.84$ | $65.8 \pm 3.74$ | $71.6 \pm 3.12$ | $61.9 \pm 3.81$ | $85.9 \pm 2.25$ | $77.5 \pm 2.82$ | $67.6 \pm 3.82$ | $73.5 \pm 3.21$ | $63.7 \pm 3.94$ |
| **HyCAS** | $\mathbf{89.5 \pm 1.64}$ | $\mathbf{88.6 \pm 2.33}$ | $\mathbf{77.3 \pm 3.14}$ | $\mathbf{86.9 \pm 2.42}$ | $\mathbf{74.4 \pm 3.33}$ | $\mathbf{91.3 \pm 2.63}$ | $\mathbf{90.4 \pm 2.82}$ | $\mathbf{79.3 \pm 3.52}$ | $\mathbf{88.2 \pm 2.63}$ | $\mathbf{76.7 \pm 3.34}$ |

(10+90 warm-up and main training), with a batch size of 300 using `SGD` as the optimizer with learning rate $10^{-1}$, momentum 0.9, and weight decay $10^{-4}$. A step scheduler is used with step size 30 and decay factor $\gamma = 0.1$. `HyCAS` injects *internal* spatial and channel attention noise on each forward pass; convolutions are regularized with spectral scaling (via `FFT` or power iteration) combined with GroupSort activations and convex residual gating, ensuring the network remains at most 2-Lipschitz. We optimize using categorical cross-entropy loss and report top-1 accuracy.

**Adversarial Training with `HyCAS` for Empirical Robustness.** Let $f_\theta : \mathbb{R}^d \to \mathbb{R}^C$ be the `HyCAS`-integrated base classifier with parameters $\theta$, mapping an input $x$ to its logits $f_\theta(x)$. For a given clean sample $(x, y)$ and perturbation budget $\epsilon$, an adversarial example $x^*$ is obtained by maximizing the loss inside the $\epsilon$-ball around $x$:

$$x^* = \underset{x^* : \|x^* - x\| \leq \epsilon}{\arg\max} \; \mathcal{L}_{\texttt{HyCAS}}\big(f_\theta(x^*; \Omega[A]), y\big), \tag{21}$$

where $\mathcal{L}_{\texttt{HyCAS}}$ is the task loss and $\Omega[A]$ emphasizes that gradients are taken in the *attack* phase.

Adversarial training then solves the following classical min–max problem:

$$\min_{\theta[I]} \; \max_{x^* : \|x^* - x\| \leq \epsilon} \; \mathcal{L}_{\texttt{HyCAS}}\big(f_\theta(x^*; \Omega[A]), y\big), \tag{22}$$

where $\theta[I]$ denotes the parameters updated during the *inference* phase.

As detailed in Algorithm 3, combining this min–max optimization with all integrated streams enables `HyCAS` to maintain strong adversarial resilience at inference time.

Table 7: Robust accuracy (%) against $\ell_\infty$ attacks (PGD-20 and AA-20) on `HAM10000` (left) and `EyePACS` (right) at $\epsilon \in \{8/255, 16/255\}$.

| Method | HAM10000 | | | | | EyePACS | | | | |
|---|---|---|---|---|---|---|---|---|---|---|
| | Clean | PGD–20 | | AA–20 | | Clean | PGD–20 | | AA–20 | |
| | | 8/255 | 16/255 | 8/255 | 16/255 | | 8/255 | 16/255 | 8/255 | 16/255 |
| AT | $75.2 \pm 2.94$ | $58.3 \pm 3.81$ | $48.4 \pm 3.75$ | $54.2 \pm 3.93$ | $44.2 \pm 3.80$ | $78.2 \pm 2.91$ | $62.4 \pm 2.72$ | $52.8 \pm 3.52$ | $58.3 \pm 2.83$ | $48.2 \pm 3.61$ |
| RPF | $74.3 \pm 2.86$ | $64.6 \pm 3.62$ | $53.3 \pm 3.52$ | $62.6 \pm 3.71$ | $50.4 \pm 3.58$ | $77.1 \pm 2.90$ | $68.5 \pm 2.61$ | $57.6 \pm 3.53$ | $66.4 \pm 2.73$ | $54.4 \pm 3.44$ |
| CTRW | $74.3 \pm 2.75$ | $64.7 \pm 3.42$ | $54.2 \pm 3.43$ | $63.3 \pm 3.48$ | $51.2 \pm 3.52$ | $76.4 \pm 2.84$ | $70.1 \pm 2.53$ | $58.5 \pm 3.43$ | $69.7 \pm 2.64$ | $56.1 \pm 3.31$ |
| DCS | $73.2 \pm 2.94$ | $63.5 \pm 3.65$ | $52.4 \pm 3.30$ | $61.4 \pm 3.72$ | $49.5 \pm 3.45$ | $76.4 \pm 2.94$ | $67.5 \pm 2.62$ | $56.2 \pm 3.35$ | $65.3 \pm 2.74$ | $53.6 \pm 3.46$ |
| ARS | $69.8 \pm 3.22$ | $56.2 \pm 3.95$ | $46.5 \pm 3.62$ | $52.7 \pm 4.10$ | $42.8 \pm 3.71$ | $72.9 \pm 3.97$ | $61.9 \pm 2.91$ | $50.3 \pm 3.61$ | $57.6 \pm 2.94$ | $46.5 \pm 3.73$ |
| DRS | $68.9 \pm 3.28$ | $55.3 \pm 3.84$ | $45.7 \pm 3.75$ | $51.6 \pm 4.12$ | $41.8 \pm 3.81$ | $71.9 \pm 3.86$ | $60.5 \pm 2.81$ | $49.6 \pm 3.82$ | $56.5 \pm 2.92$ | $45.7 \pm 3.94$ |
| **HyCAS** | $\mathbf{74.6 \pm 2.74}$ | $\mathbf{67.8 \pm 3.43}$ | $\mathbf{55.3 \pm 3.14}$ | $\mathbf{65.8 \pm 3.42}$ | $\mathbf{53.1 \pm 3.33}$ | $\mathbf{77.6 \pm 2.79}$ | $\mathbf{72.6 \pm 2.72}$ | $\mathbf{60.5 \pm 3.43}$ | $\mathbf{71.8 \pm 2.82}$ | $\mathbf{58.3 \pm 3.32}$ |

Table 8: RS/DRS vs HyCAS certified accuracy on `EyePacs NCT-CRC-HE-100K` benchmarks. The best performance under each training strategy is **bold**.

| Approach | $\sigma$ | EyePacs | | | | | NCT-CRC-HE-100K | | | | |
|---|---|---|---|---|---|---|---|---|---|---|---|
| | | $\ell_2$ radius $r$ (%) | | | | | $\ell_2$ radius $r$ (%) | | | | |
| | | 0.00 | 0.25 | 0.50 | 0.75 | 1.00 | 0.00 | 0.25 | 0.50 | 0.75 | 1.00 |
| DRS | 0.25 | $81.3 \pm 1.92$ | $60.1 \pm 2.83$ | $50.1 \pm 1.53$ | $40.9 \pm 2.04$ | $26.2 \pm 2.69$ | $89.5 \pm 2.78$ | $67.6 \pm 2.63$ | $56.7 \pm 3.35$ | $45.3 \pm 2.27$ | $30.5 \pm 3.54$ |
| | 0.5 | $78.9 \pm 0.91$ | $57.7 \pm 2.82$ | $51.8 \pm 1.34$ | $42.1 \pm 0.96$ | $30.4 \pm 3.74$ | $85.2 \pm 1.67$ | $65.6 \pm 1.89$ | $56.2 \pm 1.23$ | $48.2 \pm 1.67$ | $33.1 \pm 1.20$ |
| ARS | 0.25 | $83.1 \pm 1.35$ | $62.9 \pm 2.93$ | $47.9 \pm 1.94$ | $40.7 \pm 2.32$ | $36.2 \pm 3.73$ | $91.7 \pm 1.84$ | $69.3 \pm 2.07$ | $59.4 \pm 2.37$ | $48.3 \pm 3.91$ | $31.9 \pm 2.63$ |
| | 0.5 | $80.9 \pm 1.14$ | $60.7 \pm 2.27$ | $51.5 \pm 1.42$ | $42.2 \pm 2.68$ | $37.5 \pm 0.98$ | $87.8 \pm 2.25$ | $68.4 \pm 1.91$ | $60.1 \pm 1.01$ | $50.7 \pm 0.53$ | $34.2 \pm 1.39$ |
| HyCAS | 0.25 | $\mathbf{86.7 \pm 0.97}$ | $\mathbf{66.1 \pm 1.62}$ | $\mathbf{51.4 \pm 2.74}$ | $\mathbf{45.7 \pm 1.27}$ | $\mathbf{39.2 \pm 2.61}$ | $\mathbf{95.4 \pm 2.02}$ | $\mathbf{72.9 \pm 1.63}$ | $\mathbf{63.1 \pm 1.59}$ | $\mathbf{51.7 \pm 3.11}$ | $\mathbf{33.2 \pm 1.84}$ |
| | 0.5 | $\mathbf{82.6 \pm 1.89}$ | $\mathbf{63.9 \pm 1.74}$ | $\mathbf{53.2 \pm 2.81}$ | $\mathbf{46.3 \pm 1.45}$ | $\mathbf{41.5 \pm 1.67}$ | $\mathbf{92.3 \pm 0.61}$ | $\mathbf{71.7 \pm 1.11}$ | $\mathbf{63.4 \pm 1.36}$ | $\mathbf{52.2 \pm 1.21}$ | $\mathbf{36.9 \pm 2.57}$ |

---

**Algorithm 3** : Adversarial Training with `HyCAS`

---

1: **Require:** `HyCAS` integrated base classifier $\{_\theta(\cdot)$ with learning parameter $\theta$; Perturbation size $\epsilon$; Attack step size $a$; Number of attack iterations $k$; Training set $\{x, y\}$; Generated attention noise $M_\omega$ by `RANI` module.

2: **Procedure:**

3: **while** not converged **do**

4:     Sample a batch $\{bx, by\}_{i=1}^n$ from $\{x, y\}$;

5:     **Apply `HyCAS` for Attack phase:**

$$z(x)_{:,:,c} = \sum_{b \in \mathcal{B}} \alpha_{b,c} \left[ G_b(x; \Omega) \right]_{:,:,c} + R\Big( \sum_{b \in \mathcal{B}} \alpha_{b,c} \left[ G_b(x; \Omega) \right]_{:,:,c}; M_\omega \Big) ; \quad c = 1, \ldots, C.$$

6:     Compute the HyCAS–integrated network is optimised with a standard $\ell_\infty$ loss:

$$\mathcal{L}_{HyCAS} = \zeta \odot \mathcal{L}_{FDPAN} + \varphi \odot \mathcal{L}_{SNCAN} + \nu \odot \mathcal{L}_{RPFAN} + \kappa \odot \mathcal{L}_{RANI}$$

$$\min_\theta \max_{x^*} \left( \mathcal{L}_{\text{HyCAS}}(\{_\theta(x^*; \Omega[A]), y) \right) \quad \text{s.t. } \|x^* - x\| \le \epsilon$$

7:     **Generate Adversarial Examples:**

8:     Randomly initialize adversarial perturbation $\delta$;

9:     **for** $i = 1$ to $k$ **do**

$$\delta \leftarrow \delta + a \cdot \text{sign}\left( \nabla_{bx} \mathcal{L}_{\text{HyCAS}}(\{_\theta(bx^*; \Omega), by) \right) \quad bx^* \leftarrow \text{Clip}_{bx}^\epsilon(bx + \delta)$$

10:     **end for**

11:     **Apply `HyCAS` for Inference phase:**

$$z(x)_{:,:,c} = \sum_{b \in \mathcal{B}} \alpha_{b,c} \left[ G_b(x; \Omega) \right]_{:,:,c} + R\Big( \sum_{b \in \mathcal{B}} \alpha_{b,c} \left[ G_b(x; \Omega) \right]_{:,:,c}; M_\omega \Big) ; \quad c = 1, \ldots, C.$$

$$\min_{\theta[I]} \max_{x^*} \left( \mathcal{L}_{\text{HyCAS}}(\{_\theta(x^*; \Omega[A]), y) \right) \quad \text{s.t. } \|x^* - x\| \le \epsilon$$

12:     **Adversarial Training Optimization:**

$$\theta = \theta - \nabla_\theta \Big( \mathcal{L}_{\text{HyCAS}}(\{_\theta(x^*, \Omega), y) \Big)$$

13: **end while**

---

Table 9: Computational cost comparison between vanilla backbones and their HyCAS-integrated counterparts. We report the number of parameters (M), FLOPs (G), activation memory (MB), and inference time (ms) for ResNet-110 on CIFAR-10 ($32 \times 32 \times 3$) and ResNet-50 on ImageNet-1K ($224 \times 224 \times 3$).

| Backbone | Variant | Inputs | Parameters (M) | FLOPs (G) | Memory (MB) | Inference Time (ms) |
|---|---|---|---|---|---|---|
| ResNet-110 | Vanilla | $32 \times 32 \times 3$ | 27.6 | 8.10 | 105.33 | 0.33 |
| | HyCAS | | 57.8 | 126.5 | 220.5 | 5.14 |
| ResNet-50 | Vanilla | $224 \times 224 \times 3$ | 23.3 | 113.8 | 88.89 | 4.62 |
| | HyCAS | | 102.7 | 1682.5 | 391.9 | 68.3 |

## A.9 ADDITIONAL EXPERIMENTAL RESULTS

**Empirical robustness on Chest Xray, Histopathology, Dermoscopy, Funduscopy modalities.** Across our empirical evaluations (Tables 6–7), HyCAS achieves the highest robust top-1 accuracy under PGD-20 and AA-20 at $\epsilon \in \{8, 16\}/255$. Specifically, on the NIH-CXR benchmark, HyCAS retains robust accuracy, outperforming the leading baseline (CTRW) by up to **+1.0–2.2%** while competitive clean-set accuracy (89.5% vs. 89.1%) vs. AT. A similar trend appears on the NCT-CRC-HE-100K dataset, where HyCAS records robust accuracies of **76.7–79.3%** against the same attacks, edging past CTRW by $\approx +1\%$ and leaving earlier certified defences (e.g., ARS, DRS) more than **+7%** behind at the larger perturbation strength. Dermoscopic HAM10000 and fundus-image EyePACS exhibit the same hierarchy: HyCAS secures robust accuracies of **53.1–67.8%** against PGD-20 and AA-20 attacks on HAM10000—roughly **+1.1–7.4%** better than the next-best adversarial defence—and widens the margin on EyePACS to **58.3–72.6%**, thereby surpassing the leading baseline CTRW by **+2.0–3.3%**. *Together, these results show that HyCAS transfers its randomized Lipschitz-based strategy from certified to empirical settings, preserving clean accuracy while preserving adversarial robustness against strong first-order attacks.*

Evaluation under stronger PGD attacks on other vision benchmarks (Figs. 8–9) reveals that HyCAS not only wins at standard PGD-20 settings (App. A.8) but also sustains its lead as the adversary grows stronger. When the perturbation strength is swept from $\epsilon = 0.01 \rightarrow 0.08$ on CIFAR-10, HyCAS traces the upper envelope of robust-accuracy curves, preserving a $\approx 10\%$ *gap at the maximum perturbation strength*, where all baselines collapse sharply. An analogous pattern emerges on CIFAR-100 as the number of PGD iterations climbs from **10 → 100**: while every defense degrades monotonically, HyCAS declines more gracefully and ends **7–12%** above the closest competitor at 100 steps, confirming that its internally resampled attention noise and random projections thwart extended optimization. Thus, *this randomized, Lipschitz-constrained design* scales gracefully with both perturbation size and steps, offering adversarial robustness and a broader safety margin.

**Certified adversarial robustness on EyePacs and NCT-CRC-HE-100K.** Across the complete set of baselines in Table 8, HyCAS delivers the strongest certified accuracy for every inspected radius–noise pair. On the EyePacs benchmark, at the representative medium radius $r=0.75$ it reaches **45.7%** certified accuracy for $\sigma = 0.25$ and **46.3%** for $\sigma = 0.50$, outpacing the best competing method (DRS/ARS) by 4.1–4.8%. Even in the large-radius tail ($r=1.0$), HyCAS maintains **39.2%** ($\sigma = 0.25$) and **41.5%** ($\sigma = 0.50$), widening the gap over the strongest baseline by up to 4.0%.

A comparable pattern emerges on the NCT-CRC-HE-100K histopathology dataset. At $r=0.75$, HyCAS secures **51.7%** ($\sigma = 0.25$) and **52.2%** ($\sigma = 0.50$), improving on the best baseline by 1.5–3.4%. In the challenging $r=1.0$ regime it still records **33.2%** ($\sigma = 0.25$) and **36.9%** ($\sigma = 0.50$), extending the lead to as much as 2.7%.

Besides robustness, `HyCAS` achieves the highest clean accuracy on both datasets—**86.7%** on Eye-Pacs and **95.4%** on `NCT-CRC-HE-100K` for $\sigma = 0.25$—underscoring that its certified gains do not come at the expense of nominal performance.

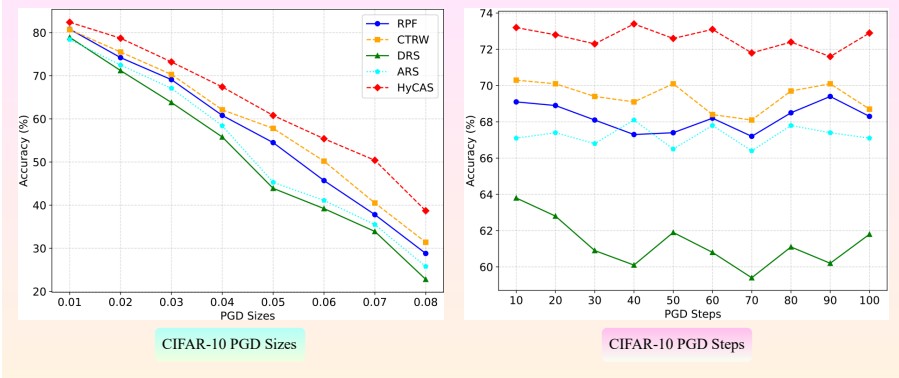

Figure 8: Empirical robustness of `HyCAS` versus leading baselines (`RPF`, `CTRW`, `DRS`, `ARS`) on `CIFAR-10` under strong `PGD` attacks. We evaluate two settings: (1) perturbation sizes $\epsilon$ from 0.01 to 0.08 and (2) iteration steps from 10 to 100.

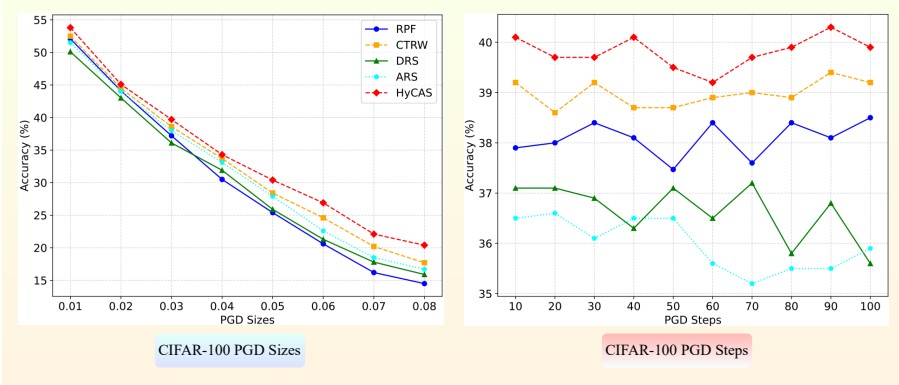

Figure 9: Empirical robustness of `HyCAS` versus leading baselines (`RPF`, `CTRW`, `DRS`, `ARS`) on `CIFAR-100` under strong `PGD` attacks. We evaluate two settings: (1) perturbation sizes $\epsilon$ from 0.01 to 0.08 and (2) iteration steps from 10 to 100.

**Computational cost analysis.** Table 9 quantifies the computational overhead of integrating Hy-CAS into standard CNN backbones. On CIFAR-10 with ResNet-110, replacing the vanilla backbone with its HyCAS counterpart roughly doubles the parameter count and activation memory (27.6M $\rightarrow$ 57.8M and 105.33MB $\rightarrow$ 220.5MB, i.e., $\approx 2.1\times$ in both cases). In contrast, FLOPs and inference time increase by about an order of magnitude: 8.10G $\rightarrow$ 126.5G FLOPs and 0.33ms $\rightarrow$ 5.14ms i.e., $\approx 15.6\times$. A similar pattern is observed on ImageNet-1K with ResNet-50, where HyCAS induces a $4.4\times$ increase in parameters and memory (23.3M $\rightarrow$ 102.7M and 88.89MB $\rightarrow$ 391.9MB), while FLOPs and inference time grow by $\approx 14.8\times$ (113.8G $\rightarrow$ 1682.5G and 4.62ms $\rightarrow$ 68.3ms).

To better localize this overhead, we also compare HyCAS with a single intermediate $3 \times 3$ convolutional block of 256 channels in ResNet-110. Substituting this standard convolution with a Hy-CAS block increases the number of parameters by $5.09\times$, whereas the corresponding FLOPs rise by $14.05\times$, indicating that most of the extra cost stems from repeated stochastic operations rather than weight storage. These block-level ratios are consistent with the backbone-level trends above, where HyCAS trades only $\approx 2$–$4\times$ more parameters and memory for roughly a $\approx 15\times$ increase in arithmetic and runtime.

Overall, HyCAS is best characterized as a parameter-moderate but compute-heavy defense: the three Lipschitz-constrained stochastic streams (FDPAN, SNCAN, and RPFAN) together with the

RANI module primarily inflate FLOPs and inference time, while absolute inference times remain in a practical range (a few milliseconds on CIFAR-10 and tens of milliseconds on ImageNet per image on an A100-class GPU). In return, HyCAS consistently delivers state-of-the-art certified and empirical robustness across CIFAR-10, ImageNet, and the medical-imaging benchmarks reported in Tables 1– 4 and 6– 8, making this overhead acceptable for many offline or near real-time deployment scenarios.

## A.10 ABLATION STUDY

Table 10 traces a controlled progression from the regularized-smoothing (RS) baseline to the full HyCAS model, revealing how each block incrementally strengthens robustness. Replacing ordinary convolutions with the spectrally-normalized **SNCAN** backbone already raises certified accuracy at the medium radius ($r$=0.75) from 32.4% to 36.9% and improves PGD-20 robustness by 3.7%, indicating that spectral control alone substantially smooths the gradient landscape. When the orthogonal **RPFAN** branch is introduced next, cer-

Table 10: Module–wise contribution for certified and empirical adversarial robustness on CIFAR-10 benchmark. Here, we have used $\sigma = 0.50$.

| Variant | Accuracy (%) | |
|---|---|---|
| | Certified ($r$=0.75) | PGD-20 ($\epsilon$=8/255) |
| RS | 32.4 | 57.5 |
| + SNCAN | 36.9 | 61.2 |
| + RPFAN | 40.2 | 64.8 |
| + FDPAN | 42.3 | 66.7 |
| + RANI | **44.3** | **70.1** |

tified and empirical accuracies climb further to 40.2% and 64.8%, respectively, showing that decorrelated projections supply complementary features beyond spectral stabilization. Extending the spectrum through **FDPAN** yields another gain—42.3% certified and 66.7% empirical—confirming that high-frequency cues remain valuable even under $\ell_2$ certification. Finally, injecting data-independent attention noise via **RANI** closes the gap between certified and empirical metrics, culminating in 44.3% certified accuracy and 70.1% PGD-20 robustness, which exactly matches the performance of the complete HyCAS system. Altogether, these sequential additions deliver an aggregate improvement of +11.9% certified and +12.6% empirical robustness over the RS baseline, underscoring that each module contributes a distinct yet additive benefit to adversarial defense.

## A.11 DISCUSSION

HyCAS is designed to reduce the gap between provable $\ell_2$ robustness and empirical $\ell_\infty$ robustness by combining a globally Lipschitz design with carefully structured internal stochasticity. This section focuses on why this design yields strong certified guarantees and how it simultaneously improves empirical robustness across natural and medical imaging benchmarks.[3]

**Certified robustness from a randomized Lipschitz network.** The certified guarantee provided by HyCAS is margin-based: if

$$Z(x) = \mathbb{E}_\Omega[s_\theta(x;\Omega)], \qquad \Delta Z(x) = Z_{(1)}(x) - Z_{(2)}(x),$$

denote respectively the logits averaged over the internal randomness and their top-two gap, and if $\text{Lip}(Z) \leq 2$, then

$$r_2(x) = \frac{\Delta Z(x)}{4}$$

is a valid pointwise $\ell_2$ certificate (Corollary 1). This guarantee acts on the *expected logits* of a globally $\leq$ 2-Lipschitz network obtained by stacking HyCAS blocks, each of which combines a 1-Lipschitz deterministic core (spectrally normalized convolutions, orthogonal channel mixing, low-pass DCT, and spectrally normalized random projections) with a 2-Lipschitz attention-noise residual. The convex fusion of the three streams and the expectation over $\Omega$ preserve the global $\leq$ 2-Lipschitz constant.

---

[3]See Sections 3–4 and Appendix A for full details of the architecture, certification scheme, and experimental setup.

This certificate is qualitatively different from the randomized smoothing (RS) radius of Cohen et al. based on smoothed class probabilities and Gaussian concentration. It is not a relaxed or "looser" version of the RS bound: RS certifies the majority vote of a noise-perturbed classifier, whereas HyCAS certifies the margin of a Lipschitz-constrained expected-logit map. Which radius is larger in practice therefore depends on how training shapes the *margin distribution* under each mechanism, not on a direct comparison of constants in the formulas.

Empirically, HyCAS consistently achieves higher certified accuracy than RS, IRS, DRS, ARS and deterministic Lipschitz baselines (LOT, SLL) at all reported radii on CIFAR-10/100, ImageNet and the medical datasets (Tables 1- 2, 8). For example, on CIFAR-10 at radius $r = 0.75$ and $\sigma \in \{0.25, 0.50\}$, HyCAS attains $44.3\%$ certified accuracy, which is 5.2–18.2 points above prior methods; at $r = 2.0$ and $\sigma = 0.50$, it still retains $12.5\%$, exceeding the strongest baseline by 4.0– 12.5 points. Similar gains appear on ImageNet, CelebA, HAM10000, NIH-CXR, EyePACS and NCT-CRC-HE-100K. These improvements cannot be explained by small fluctuations in clean accuracy alone, indicating that the architecture and training jointly enlarge $\Delta Z(x)$ for many points while respecting the conservative $\mathrm{Lip}(Z) \leq 2$ envelope. For example, on CIFAR-10 with $\sigma = 0.25$, RS attains $75.3\%$ smoothed clean accuracy at $r = 0$ and $26.1\%$ certified accuracy at $r = 0.75$, whereas HyCAS reaches $85.4\%$ at $r = 0$ and $44.3\%$ at $r = 0.75$ (Table 1). Thus, the gain at a non-zero radius (+18.2 points at $r = 0.75$) is substantially larger than the gain at $r = 0$ (+10.1 points), indicating that HyCAS's Lipschitz-constrained hybrid architecture enlarges robust margins around inputs rather than merely improving accuracy on unperturbed data.

The ablation in Table 10 illustrates this mechanism: starting from an RS-style baseline, replacing standard convolutions by SNCAN, then adding RPFAN, FDPAN and finally RANI, monotonically increases certified accuracy at $r = 0.75$ on CIFAR-10 from $32.4\%$ to $44.3\%$. Each variant uses the same backbone, noise level and objective; the only changes are architectural. This progression shows that the larger certified radii arise from reshaping the margin distribution under a Lipschitz constraint, rather than from a fundamentally stronger analytical bound.

From a theoretical perspective, the contribution is to extend margin-based $\ell_2$ certification to networks with *internal stochasticity* and to demonstrate that such networks can be trained at scale. The analysis proves that the expected logits of a HyCAS network remain $\leq 2$-Lipschitz despite random projections and stochastic attention, and that this property can be enforced layerwise (via spectral normalization and calibrated residual scaling) in standard CNN backbones while still achieving competitive clean accuracy on large benchmarks.

**Mechanisms underlying empirical $\ell_\infty$ robustness.** The same ingredients that support the certificate also improve robustness against strong $\ell_\infty$ attacks such as APGD, PGD and AutoAttack. Several aspects of the design are central:

- **Spectral control.** SNCAN replaces standard convolutions by spectrally normalised ones, constraining the operator norm of each kernel and smoothing the loss landscape. This reduces the ability of first-order attacks to exploit sharp directions in the input space.

- **Random projections.** RPFAN combines an orthogonal $1 \times 1$ channel pre-mix with batch-aware spectral normalisation of random projection filters. This decorrelates channels and redistributes energy while preserving local geometry, making adversarial search less aligned with a single vulnerable feature direction.

- **Frequency-aware filtering.** FDPAN uses low-pass DCT masking and orthogonal mixing to suppress brittle high-frequency content where small $\ell_\infty$ perturbations can hide, without discarding all high-frequency information that remains useful for classification.

- **Randomized Attention Noise Injection (RANI).** RANI injects a bounded, data-independent attention mask after each Lipschitz core and at the fused output. For each fixed noise realization, the module is 2-Lipschitz, but across evaluations it presents a shifting, yet certifiably bounded, optimisation landscape.

Combined with adversarial training (Algorithm 3), these components yield strong empirical robustness. Across all four medical benchmarks, HyCAS attains the highest robust accuracy under APGD-20 and AA-20 at $\epsilon \in \{8/255, 16/255\}$ while maintaining clean accuracy that is on par with

or slightly better than existing adversarially trained and randomized baselines (Tables 3- 4, 6- 7). On CIFAR-10/100, HyCAS dominates the robust-accuracy curves across perturbation sizes and attack steps (Figures 2- 3, 8- 9): as $\epsilon$ increases from $0.01$ to $0.08$ or the number of PGD/APGD iterations grows from $10$ to $100$, all baselines degrade sharply, whereas HyCAS declines more gradually and preserves a 7–12 point margin at the strongest settings.

Importantly, during attack generation the internal noise is held fixed per adversarial example, so gradients remain well-defined; stochasticity is only exploited across examples, not within the optimisation path. Together with the global Lipschitz control, this suggests that the improved $\ell_\infty$ robustness comes from genuinely harder optimisation and smoother gradients rather than from gradient masking.

The ablation in Table 10 again mirrors this: each successive module added to the RS baseline improves both certified accuracy and PGD-20 robustness on CIFAR-10 at $r = 0.75$ and $\epsilon = 8/255$, culminating in a total gain of $+11.9\%$ certified and $+12.6\%$ empirical robustness. This tight coupling supports the view that HyCAS does not trade certified and empirical robustness against each other, but instead uses a shared Lipschitz–randomized structure to improve both.

**Certified–empirical trade-offs.** Figure 4 summarizes HyCAS as a three-way Pareto frontier that trades clean accuracy, certified $\ell_2$ accuracy at radius $r$, and empirical $\ell_\infty$ robustness at perturbation strength $\epsilon$. The frontier is smooth and strictly decreasing: enlarging the certified radius inevitably reduces empirical robustness. Two systematic gaps appear.

First, in the small-perturbation regime, empirical $\ell_\infty$ robustness lies well above the certified $\ell_2$ accuracy at comparable scales, reflecting the inherent pessimism of worst-case Lipschitz bounds. Many points are robust in practice beyond what a global constant can certify. Second, for large radii and perturbations, the gap widens further due to the norm mismatch: the inequality $\|\delta\|_2 \leq \sqrt{d}\,\|\delta\|_\infty$ is loose at image scale, so a model that is provably stable to moderate $\ell_2$ perturbations can empirically withstand much stronger $\ell_\infty$ attacks than suggested by the $\ell_2$ certificate.

Adjusting the smoothing noise $\sigma$ provides a practical knob along this frontier. Increasing $\sigma$ from $0.25$ to $0.50$ leaves performance at intermediate radii (e.g., $r = 0.75$) almost unchanged, yet substantially boosts certified accuracy in the high-radius tail: on CIFAR-10, accuracy at $r = 2.0$ increases from $8.5\%$ to $12.5\%$, and on ImageNet from $5.4\%$ to $24.8\%$, while small-$\epsilon$ APGD-20 robustness remains competitive. This behaviour shows that, despite the conservative constant $1/4$ in the margin bound, training under a global $\mathrm{Lip} \leq 2$ constraint can still produce margin distributions that deliver non-trivial certified radii without destroying empirical robustness.

**Limitations.** The theoretical guarantees in this work are derived from a global $\leq 2$ Lipschitz envelope and a margin bound $r_2(x) = \Delta Z(x)/4$ on the expected logits. This certificate is based on different assumptions than randomized-smoothing bounds, which operate on smoothed class probabilities, and the two guarantees are therefore not directly ordered in terms of tightness. Our analysis does not attempt to prove that the resulting radius is universally stronger than the randomized-smoothing radius; instead, it shows that, for the randomized, Lipschitz-constrained HyCAS architecture, shaping the margin distribution under a global $\leq 2$-Lipschitz constraint yields practically useful $\ell_2$ radii that empirically improve on RS-style baselines using the same backbones. The present theory is restricted to $\ell_2$ perturbations, while robustness to $\ell_\infty$ attacks is assessed empirically, and extending the framework to tighter, norm-adaptive or direct $\ell_\infty$ certificates is left for future work. Finally, HyCAS is compute–heavy: integrating three stochastic streams and the RANI module into standard CNN backbones roughly doubles parameters and memory but increases FLOPs and inference time by about an order of magnitude (Table 9). Consequently, our method is most suitable for offline or near real-time scenarios where this overhead is acceptable, and designing lighter-weight HyCAS variants is an important direction for future work.

