# OpenReview forum: "CERTIFIED VS. EMPIRICAL ADVERSARIAL ROBUSTNESS VIA HYBRID CONVOLUTIONS WITH ATTENTION STOCHASTICITY"
_ICLR.cc/2026/Conference — ICLR 2026 Poster_

### Official Review · Reviewer_roSN · 2025-10-22

**Soundness:** 4
**Presentation:** 4
**Contribution:** 3
**Rating:** 8
**Confidence:** 4

**Summary:**

This paper introduces an architectural modification to standard CNNs that makes them more robust to (particularly $\ell_\infty$) adversarial attacks. HyCas has 3 components, each of which are 2-Lipschitz, and are fused via a convex operation, which allows them to obtain an $\ell_2$ robustness certificate. There are some works that also introduce 1-Lipschitz or spectrally normalized CNNs, but I find the internal stochasticity introduced in HyCas (via random projections, random attention noise), to be principled and interesting, and also standing out as novel in comparison to prior work. I also find the work to be refreshing from the perspective of tackling the adversarial robustness problem in a new way, as opposed to yet another method for solving the vanilla $\ell_2$/$\ell_\infty$ robustness problem. Experiments convincingly demonstrate the practical utility of the proposed method.

**Strengths:**

-- A new architecture, with several novel components, which are both well motivated and nicely executed

-- Bridges a key gap between certified and empirical robustness

-- Provide formal proofs that the network is 2-Lipschitz, which to my reading are convincing

-- Benchmark on both natural and medical image datasets

-- Comprehensive experiments in terms of number of defenses compared to

**Weaknesses:**

-- The three parallel streams plus convex gating presumably increase parameter count and compute cost. But the authors do not report FLOPs or latency.

-- There is no comparison to some new methods e.g. TRADES [1] or HR [2].

[1] https://arxiv.org/pdf/1901.08573
[2] https://arxiv.org/abs/2303.02251

**Questions:**

-- What is the parameter and FLOP overhead relative to a standard ConvNet block?

-- Are the certificates costly to obtain (i.e. via Monte-Carlo)?

-- Have the authors considered salience maps to visualize the sensitivity to input perturbations as opposed to standard l2 robustness methods. I think it would be interesting to see as an additional insight.

---

> ### Author Response · Authors · 2025-11-23
> **Author Response (Part 1)**
>
> We would like to express our sincere gratitude to reviewer **roSN** for the exceptionally careful reading of our manuscript and the thoughtful, detailed evaluation of HyCAS. We are **particularly grateful** for the reviewer’s clear summary of our main contributions—namely, the *introduction of an architectural modification to standard CNNs* that enhances robustness, especially to $\ell_\infty$ adversarial attacks; the design of HyCAS as a composition of *three 2-Lipschitz components fused via a convex operation*, which enables an *$\ell_2$ robustness certificate*; and the recognition that the **internal stochasticity** we introduce (through **random projections and random attention noise**) is a *principled* and **novel feature that distinguishes our approach from prior 1-Lipschitz** and **spectrally normalized CNNs**.
>
> We also deeply appreciate the reviewer’s view that our work *offers a refreshing way of tackling adversarial robustness* beyond “yet another” $\ell_2$ / $\ell_\infty$ defense, and that the experiments convincingly demonstrate the **practical utility of the proposed method**.
>
> We are very encouraged by the reviewer’s scores (**Soundness: 4 – excellent, Presentation: 4 – excellent, Contribution: 3 – good**) and by the generous acknowledgment of the strengths of our work: the **proposal of a new architecture with well-motivated and novel components**; the effort to **bridge a key gap between certified and empirical robustness**; the **provision of formal proofs** that the network is **2-Lipschitz**, which the reviewer finds convincing; the **evaluation on both natural and medical image datasets**; and the **comprehensive experimental comparison against a broad set of defenses**.
>
> The reviewer’s insightful and constructive comments have been extremely valuable in improving the clarity, rigor, and presentation of the paper. Below, we respectfully address each of the points raised by reviewer **roSN** in detail.
>
> **Note: Weakness denotes as W and question represents as Q; Edits are colored in blue in the revised paper.**
>
> **W1/Q1 (Computational overhead of additional modules).** We agree that reporting the *computational cost of the additional HyCAS modules* is important. In the revised manuscript we now explicitly quantify *parameters*, *memory size*, *FLOPs*, and *inference time* on both **CIFAR-10** and **ImageNet** benchmarks, and we discuss these numbers in the main text and appendix.
> *Backbone-level overhead (parameters, memory, FLOPs, inference time)*.
>
> We added **Table 9** in Appendix A.9 **(page 26, lines 1350–1360)**, which compares vanilla ResNet backbones with their *HyCAS-integrated counterparts in terms of number of parameters (M), FLOPs (G), memory size (MB), and Inference time (ms).*
>
> Concretely, on **CIFAR-10** / ResNet-110, HyCAS roughly **doubles parameters** and **memory size** (**27.6M** → **57.8M**; **105.33MB** → **220.5MB**, ≈2.1×), while **FLOPs** and **inference time** increase by about 15.6× (**8.10G** → **126.5G**; **0.33ms** → **5.14ms**). On **ImageNet-1k** / ResNet-50, HyCAS induces a 4.4× increase in **parameters** and **memory size** (**23.3M** → **102.7M**; **88.89MB** → **391.9MB**) and about 14.8× more **FLOPs** and runtime (**113.8G** → **1682.5G**; **4.62ms** → **68.3ms**). These details are summarized and interpreted in the new *“Computational cost analysis”* paragraph **(page 27, lines 1441–1462)**.
>
> **Where the overhead comes from (block-level analysis).**
> To localize the cost to the new modules, we also compare a *standard 3×3 conv block* in ResNet‑110 with a *single HyCAS block* **(same position, 256 channels)**.
>
> As reported on **page 27, lines 1450–1456**, *substituting the convolution with a HyCAS block* increases **parameters** by **5.09×** but **FLOPs** by **14.05×**, indicating that the main overhead comes from repeated *stochastic operations* (three streams + RANI) rather than weight storage. These block-level ratios are consistent with the backbone-level trends in **Table 9**.
>
> **Practical inference times and positioning in the paper.**
> We further clarify that, despite the ≈15× relative increase, absolute inference times remain in a practical range—a few milliseconds per image CIFAR-10 and tens of milliseconds on ImageNet on an A100-class GPU **(page 28, lines 1458–1462)**.
>
> To make the **trade-off explicit** in the main narrative, we added a sentence in the *Limitations paragraph* stating that *HyCAS is compute-heavy but parameter-moderate, roughly doubling parameters and memory while increasing FLOPs and inference time by about an order of magnitude, and is therefore most suitable for offline or near real-time scenarios* **(page 30, lines 1612–1616)**.
>
> *Overall, we hope this clarifies that the revised paper now fully reports and discusses the memory and inference-time overhead of the HyCAS modules at both backbone and block level, directly addressing the concern in* **W1/Q1**.

---

> ### Author Response · Authors · 2025-11-23
> **Author Response (Part 2)**
>
> **W2 (Comparison)**: We sincerely thank the reviewer for this valuable suggestion. Following your recommendation, we have conducted a detailed performance comparison of HyCAS against the prior methods TRADES [1] and HR [2]. The corresponding results are presented in Figures 2–3 **(see pp. 8–9; lines 404–418, 437–452)** and are further analyzed in Section 4.3, Empirical Adversarial Robustness **(see pp. 9–10; lines 476–498)**. Across the evaluated settings, these baselines do not outperform HyCAS in terms of adversarial robustness. We attribute this improvement to HyCAS’s combination of a Lipschitz-aware architectural design with injected stochasticity, which jointly enforces stability properties and realizes an effective randomized defense. This dual mechanism enhances the capability of HyCAS to achieve stronger adversarial robustness compared to existing defenses.
>
> **Q2 (Cost of Monte‑Carlo certificates)**: Yes, we explicitly acknowledge that our Monte‑Carlo–based certification is compute‑intensive, since each certificate requires many stochastic forward passes of our heavyweight HyCAS architecture designed for state‑of‑the‑art adversarial robustness.
>
>
>
>
>
>
> [1] Hongyang Zhang, Yaodong Yu, Jiantao Jiao, Eric Xing, Laurent El Ghaoui, and Michael Jordan. Theoretically principled trade-off between robustness and accuracy. In International conference on machine learning, pp. 7472–7482. PMLR, 2019.
>
> [2] Amine Bennouna, Ryan Lucas, and Bart Van Parys. Certified robust neural networks: Generalization and corruption resistance. In International Conference on Machine Learning, pp. 2092–2112. PMLR, 2023.

---

> > ### Comment · Reviewer_roSN · 2025-11-26
> >
> > Dear Authors,
> >
> > Thank you for the response. I remain positive on the paper, and appreciate you addressing my concerns. I am also happy to see significant effort has been made to incorporate concerns of the other reviewers.

---

> > > ### Author Response · Authors · 2025-11-26
> > > **Great Appreciate for your Recognition**
> > >
> > > Dear Reviewer roSN,
> > >
> > > Thank you very much for your recognition and encouraging response. We are very glad that our clarifications have successfully addressed your concerns, and we sincerely appreciate your steady support and positive view of our work.
> > >
> > > We are also grateful for your constructive feedback and for your updated assessment of our submission. We truly appreciate the time and care you have invested in re‑reading the paper and in recognising the revisions we have made so far in response to all three reviews.
> > >
> > > Thank you again for your valuable time and kind support.

---

### Official Review · Reviewer_yF4E · 2025-10-27

**Soundness:** 2
**Presentation:** 1
**Contribution:** 2
**Rating:** 0
**Confidence:** 3

**Summary:**

This paper explores a new adversarial defense. The main contribution of the paper is to narrow the gap between the work that has been done for certified defenses under the l2 norm and empirical defenses that use the l-inf norm. This is accomplished through the use of Hybrid Convolutions with Attention Stochasticity, a new method proposed in this paper. Experimentally the results of the new method are shown on a wide range of datasets including, CIFAR-10/100, ImageNet-1K and NIH Chest X-ray.

**Strengths:**

The paper has an interesting approach to adversarial robustness and the experimental results are comprehensive in terms of the datasets used.

**Weaknesses:**

=In the abstract the metrics mentioned don’t give any specific dataset. A 7.3% better robustness is achieved with respect to which dataset?

=I would avoid the use of the word harmonize in the abstract. It is not clear what you mean when you say you harmonize provable and empirical adversarial robustness. Robustness is a measurement. If I said that I was going to harmonize kilometers and miles, would you understand what that meant? Obviously not.

“the underlying randomness in these defences is static or easily inferred once seeds are fixed, rendering them vulnerable to adaptive attacks and offering no formal guarantees.”

=In the introduction this is a very bold claim and no citations are given to back up this statement. Why would seeds be fixed? Why is the underlying randomness considered static? This is not at all clear. I would need to see several citations in the literature to back up such arguments. Otherwise it is just conjecture.

=I don’t understand why you use the term “natural” and “medical” imaging domains to describe the datasets. I know CIFAR-10, CIFAR-100, ImageNet type of datasets are images and so are chest x-rays. It sounds like you are just trying to make the empirical work more impressive when all you did is test on image datasets. You should really remove this terminology natural/medical from the paper.

=For the experiment for Figure 2, I am not convinced by the results. PGD is a very old attack. For credibility the authors at least need to update to APGD: https://arxiv.org/pdf/2003.01690

Also as far as I can tell, figure 2 is actually never referenced in the main body of the paper. Why do you have experimental results with no further explanation?

=I think the terminology is confusing when you mention that your technique is a hybrid deterministic-stochastic defense. In theory if you add randomness to a deterministic defense, we would call that defense a randomized defense. In this case your paper is combining both deterministic and stochastic defenses, but this would then mean it is stochastic. E.g., any time a deterministic defense includes randomization, it is no longer random. I am concerned that mixing such terms as you do in your paper will really confuse readers.

=Experimentally I don’t understand why so much space is wasted testing on PGD. In security we care about the strongest possible attacker. At this point PGD has widely been accepted as inferior to APGD. Therefore, there is NO reason to report PGD results in the main body of the paper. A huge  amount of space could be saved, simply by pushing all PGD results to the appendix.

=I notice certain experiments are referenced in the main body but only shown in the appendix. E.g. Figure 7. I don’t think this follows the rules because it forces reviewers to look at appendix material when we should only be considering the main body.

My overall opinion of the paper is that the experimental results offer a marginal improvement at best. However, the use of extremely convoluted writing techniques throughout the paper (natural/medical), “harmonize” and mixing up what is really a new stochastic defense that they call a nonsensical term “deterministic-stochastic”, all lead me to strong reject. I think the authors should resubmit only after major revisions are done to the writing and terminology of the paper.

**Questions:**

1. Why is the terminology so misused and convoluted? Can you fix the writing of the paper?
2. Why are you referencing results that only appear in the appendix?
3. Why did you not use APGD for all experiments instead of PGD?
4. How can you claim that your defense that has randomization is deterministic-stochastic?
5. Can you remove all PGD experiments from the paper and appendix and replace them with APGD?

---

> ### Author Response · Authors · 2025-11-23
> **Author Response (Part 1)**
>
> We would like to express our sincere gratitude to reviewer **yF4E** for the careful reading of our manuscript, the valuable and constructive feedback, and the positive assessment of HyCAS as an interesting approach to adversarial robustness with comprehensive experiments across multiple datasets. The reviewer’s insightful comments have been highly beneficial in refining the presentation, strengthening the analysis, and improving the overall clarity and quality of the paper. Below, we respectfully address each of the points raised by reviewer **yF4E** in detail.
>
> **Note: Weakness denotes as W and question represents as Q; Edits are colored in blue in the revised paper.**
>
> **W1 (Abstract metrics)**: Thank you for pointing out that the abstract was not explicit enough about what “up to 7.3%” refers to:
> The 7.3% gain refers to the certified accuracy on NIH Chest X-ray (NIH-CXR) at ℓ₂ radius r = 1.0 and smoothing noise σ = 1.0, where HyCAS achieves 41.4% certified accuracy vs. 34.1% for ARS, the strongest certified baseline in this setting, i.e. a +7.3 point improvement. This is reported in Table 2 (“Certified accuracy (%) of HyCAS and prior defenses on CelebA, HAM10000, and NIH CXR”), NIH CXR column, r = 1.0, σ = 1.0.
>
> To make this explicit, we have revised the abstract **(see Page-1; line: 24)** to name the datasets and attach the gains to them:
>
> “Extensive experiments on diverse benchmarks—including CIFAR-10/100, ImageNet-1k, NIH Chest X-ray, and HAM10000—show that HyCAS surpasses prior leading certified and empirical defenses, boosting certified accuracy by up to ≈7.3% (on NIH Chest X-ray) and empirical robustness by up to ≈3.1% (on HAM10000), without sacrificing clean accuracy.”
>
> **W2 (harmonize)**: We agree that “harmonize” is vague here and can be misinterpreted. Our intention was simply to say that **HyCAS simultaneously improves**
>
> • certified ℓ₂ robustness (via a ≤2 Lipschitz certificate on the expected logits), and
>
> • empirical ℓ∞ robustness (under strong white box attacks such as APGD/PGD/AutoAttack),
>
> while maintaining high accuracy.
>
> In the revised abstract (**See page-1; lines: 26-27**) we therefore remove “harmonize” entirely and replace it with precise wording, for example:
>
> “These results show that a randomized Lipschitz constrained architecture can **simultaneously improve** both certified ℓ₂ and empirical ℓ∞ adversarial robustness, thereby supporting safer deployment of deep models in high-stakes applications.”
>
> We have also ensured that similar vague phrasing does not appear elsewhere in the paper. Edits are colored in blue.
>
> **W3 (bold claim)**: We agree that the sentence in question (“the underlying randomness in these defences is static or easily inferred once seeds are fixed…”) was too strong and imprecise. Our intention was not to claim that prior work literally fixes a single random seed, but rather to highlight that several empirical randomized defences we compare against (CTRW, RPF) inject input-independent randomization: at inference they draw fresh noise or random weights from Gaussian distributions whose functional form and parameters are pre-specified or learned once and then reused across test inputs.
>
> In such settings, prior work has repeatedly shown that non-certified randomized defences can often be circumvented by adaptive attacks that explicitly model or average over their internal noise, e.g., via Expectation-over-Transformation and related techniques (Athalye et al., 2018; Tramer et al., 2020).
>
> In the revised manuscript we therefore **removed the “static / seeds are fixed”** wording from the introduction and replaced it with the following more precise, cited statement in the Related Work section **(see pages: 2-3, lines: 104-109)**:
>
> **“Despite these gains, these empirical defences provide no certified worst case guarantees (Yang et al., 2022; Liu et al., 2021), and many rely on input independent randomization (He et al., 2019; Jeddi et al., 2020; Ma et al., 2023; Dong & Xu, 2023); these non-certified randomized defences have often been circumvented by adaptive attacks that explicitly average over the internal noise (Athalye et al., 2018; Tramer et al., 2020).”**
>
> **W4 (“natural” and “medical” imaging domains)**: We apologise for the impression this wording created. Our intention was simply to distinguish between
>
> (i) standard vision benchmarks (CIFAR-10/100, ImageNet-1k, CelebA) and
> (ii) clinical imaging datasets (NIH Chest X-ray, NCT-CRC-HE-100K, EyePACS, HAM10000),
> and to highlight that HyCAS is evaluated on both groups.  We chose to test on clinical datasets because they are essential for real-world applicability and for addressing high-stakes decision-making in medical diagnoses. The general-purpose and domain-specific datasets differ in resolution, texture statistics, class imbalance, and deployment setting, and Tables 1–4 show that, among the methods we compare, HyCAS is the only defence that is both certified and empirical and reported on all of them.

---

> ### Author Response · Authors · 2025-11-23
> **Author Response (Part 2)**
>
> That said, we agree that repeatedly saying “natural/medical imaging domains” is stylistically heavy and can sound like marketing. In the revision we have therefore **substantially reduced this terminology**: we now refer primarily to the concrete dataset names, and only a few times to “standard vision benchmarks and medical datasets” when summarising evaluation scope. This wording is meant only to clearly describe the breadth of the evaluation, not to present the natural/medical distinction itself as a main contribution. **See pages: 1, 2, etc. and lines: 16, 56, 60, 107 etc.**
>
>
>
>
>
> **W6 / Q4 (Terminology “hybrid deterministic–stochastic”)**: We agree with the reviewer’s theoretical point: once a defense uses **randomness at inference, the overall mapping (x $\mapsto$ y) is randomized** and must be modeled as such in the threat model.
>
> Our original phrase *“hybrid deterministic–stochastic defense”* was intended only to describe the **architecture** (a deterministic Lipschitz-constrained stream plus stochastic modules), not to suggest that the defense is partly deterministic from the attacker’s point of view. We recognise that this wording was confusing and have revised the paper accordingly:
>
> * In the **abstract, introduction, and method section** we now describe HyCAS as “a **randomized defense** whose architecture combines a **deterministic Lipschitz-constrained design** with **stochastic smoothing modules** (random-projection and attention-noise mechanisms).” **See pages: 1, 2, 3, 4, 6, 10; lines: 19, 60, 68, 114-115, 194, 302-304, 487, 498, 537.**
>
> * We now use **“deterministic” only for internal building blocks** (spectrally normalized convolutions, orthogonal transforms) and **“stochastic” only for the RPF and RANI components**. The overall HyCAS network is consistently referred to as a *randomized defense*.
>
> * We also **clarify the threat model and certificate**: attacks are adaptive and know the distribution of all internal randomness, and our $\ell_2$ guarantee is stated as a *randomized Lipschitz margin certificate for the expected logits*
>
>
>
>
> $Z(x) = \mathbb{E}\omega[s{\theta}(x;\omega)]$ (Corollary 1 in the revised manuscript). **See page: 4; lines: 194-206.**
>
>
>
> With these changes, we no longer describe the defense itself as **“deterministic–stochastic”**; HyCAS is explicitly treated as a **randomized defense**, while the deterministic/stochastic terminology is used only to explain how the Lipschitz-constrained streams and stochastic modules are combined inside the architecture.
>
>
>
> **W5 / W7 / Q3 / Q5 (PGD vs. APGD)**: Thank you for raising these points. In the original submission we used PGD mainly for:
>
> 1. **Comparability** with earlier **randomized defenses (e.g., RPF [1], CTRW [2])**, which largely report PGD-based robustness; and
>
> 2. **Stress-testing behavior** as ε and the number of iterations grow, a standard way to visualize robustness decay.
>
> We agree that APGD / AutoAttack is now the stronger and more standard choice, and have revised the paper accordingly:
>
> * **APGD as the primary attacker.** All main empirical robustness tables (Tables 3–4) now report performance under ℓ∞ **APGD-20** and **AutoAttack (AA-20)** at ε ∈ {8/255, 16/255}; the evaluation protocol in Sec. 4.1 has been updated to reflect this. **See Pages: 8, 9-10; lines: 378-398, 476-488**
>
> For example,
>
> Across our empirical evaluations **(Tables 3–4)**, HyCAS achieves the highest robust top-1 accuracy under **APGD-20** and **AA-20** at ϵ ∈ {8, 16}/255. Specifically, on the **NIH-CXR** benchmark, **HyCAS** retains robust accuracy, outperforming the leading baseline (CTRW) by about +1.8–4.2% across these attacks while maintaining similar clean-set accuracy (89.5% vs. 88.4%).
>
> A similar trend ap- pears on the **NCT-CRC-HE-100K** dataset, where **HyCAS** records robust accuracies of 76.7–79.3% at ϵ = 16/255 against the same attacks, exceeding CTRW by roughly +1.5–2.6% and leaving earlier certified defences (e.g., ARS, DRS) more than +12% behind at this stronger perturbation level.
>
> Dermoscopic HAM10000 and fundus-image EyePACS exhibit the same hierarchy: HyCAS secures robust accuracies of 53.1–67.8% against APGD-20 and AA-20 attacks on HAM10000—around +1.9–3.1% better than the next-best adversarial defence—and widens the margin on EyePACS to 58.3–72.6%, thereby surpassing the leading baseline CTRW by approximately +2.1–2.8%.
>
> **Together, these results show that HyCAS transfers its randomized Lipschitz strategy from certification to empirical regimes, maintaining clean accuracy while achieving state-of-the-art adversarial robustness.**

---

> ### Author Response · Authors · 2025-11-23
> **Author Response (Part 3)**
>
> * **Figures 2–3 and text in Sec. 4.3.** **See Pages: 8, 10; lines: 404-419, 437-452, 489-498**
>
> For example,
>
> Evaluation under **stronger APGD** attacks on other **vision benchmarks** (Figs. 2–3) reveals that HyCAS not only wins at standard APGD-20 settings (App. A.8) but also sustains its lead as the adversary grows stronger. When the perturbation strength is swept from ϵ = 0.01 → 0.08 on CIFAR-10, HyCAS traces the upper envelope of robust-accuracy curves, preserving a ≈ 10% gap at the maximum perturbation strength, where all baselines collapse sharply. An analogous pattern emerges on CIFAR-100 as the number of APGD iterations climbs from 10 → 100: while every defense degrades monotonically, HyCAS declines more gracefully and ends 7–12% above the closest competitor at 100 steps, confirming that its internally resampled attention noise and random projections thwart extended optimization. **Thus, this randomized, Lipschitz-constrained design scales gracefully with both perturbation size and steps, offering adversarial robustness and a broader safety margin.**
>
> Hence, **Figures 2–3** now show robustness under **stronger APGD attacks** as ε or the number of iterations increases, and Sec. 4.3 explicitly discusses these curves (“Evaluation under **stronger APGD attacks** on other vision benchmarks (Figs. 2–3) …”), so they are no longer **“unexplained”** experimental plots.
>
> * **PGD relegated to supplementary results.**
>
> Any remaining **PGD**-based results **(e.g., Tables 6–7, Figs. 8-9 can be found in Appendix A.9)** are now clearly labelled as *supplementary results for comparison with prior baselines*. See Pages: 24-27; lines: 1264-1273, 1296-1305, 1369-1391, 1407-1438.
>
> **These changes ensure that all central robustness claims in the main paper are backed by state-of-the-art **APGD / AutoAttack** evaluations, in line with your valuable recommendation.**
>
>
>
> **W8 / Q2 (References to appendix-only material)**
> We thank the reviewer for pointing this out. In the original (previous) submission, our figure numbering and cross-references were confusing and could give the impression that some key experiments appeared only in the appendix (in particular the reference to “Figure 7”). In the revised manuscript we have reorganized things as follows:
>
> (i) **Main claims in the main body.** All quantitative experimental results that support our central claims now appear in the 10-page main paper (**Tables 1–4 and Figures 1–4**). **See Pages: 5-6, 8, 3, 9, 10; lines: 216-232, 270-282, 378-398, 118-135, 404-419, 437-452, 499-513.**
>
> (ii) **Appendix as supporting material.** The appendix is reserved for supporting content: pseudocode, proofs, detailed architectural figures (e.g., **the RPFAN diagram in App. A.5, Fig. 7**), and additional experiments and ablations (**Apps. A.9–A.10**). **See Pages: 19, 24-25; lines: 988-1017, 1242-1263.**
>
> (iii) **Explicit cross-references.** Whenever we refer to appendix material from the main text (during initial paper submission), we now do so explicitly and locally (during rebuttal with revised paper submission), e.g., “pseudocode is provided in Appendices. A.7-A.8 (Algorithms 1–3)”, “see App. A.3 for the proof of Proposition 4”, or “see App. A.5 (Fig. 7) for the detailed RPFAN diagram”. **See pages: 3-7; lines: 160-161, 179-180, 191, 214, 262, 313, 323, 366-367, etc.**
>
>
>
>
> **W9 (overall opinion)**: We appreciate the reviewer’s candid summary. We agree that the initial submission suffered from confusing terminology, and we have made substantial revisions to address this regarding all **weaknesses and questions**. HyCAS is now consistently described as a **randomized defense** whose architecture combines a deterministic Lipschitz‑constrained design with stochastic smoothing modules, vague phrases such as “harmonize provable and empirical robustness” have been removed, and the “natural/medical” wording has been greatly reduced in favour of explicit dataset names.
>
> Regarding the concern that the empirical gains are only marginal, we would like to clarify the magnitude and consistency of the improvements, based on the **revised Tables 1–4 and Sec. 4.2–4.3**:
>
> • **Certified ℓ₂ robustness on CIFAR‑10 / ImageNet.**
>   In Table 1, HyCAS attains the best certified accuracy on every (r, σ) pair. For example, on CIFAR‑10 at the representative medium radius r = 0.75, HyCAS reaches 44.3% certified accuracy, improving on the strongest baseline by 5.2 points. Even at r = 2.0, it maintains 12.5% certified accuracy versus 8.5% for the best baseline (+4.0 points).
>
> On ImageNet, in the large‑radius regime (r = 1.5), HyCAS attains 32.7% certified accuracy and exceeds all baselines by between roughly 2 and 30 points, while also delivering the highest clean accuracy on both datasets (85.4% on CIFAR‑10, 72.3% on ImageNet).

---

> ### Author Response · Authors · 2025-11-23
> **Author Response (Part 4)**
>
> **• Certified robustness on face and Skin, Chest Xray datasets.**
> Table 2 shows a similar pattern on CelebA, HAM10000 and NIH‑CXR: HyCAS improves certified accuracy by about 4–18 points depending on the dataset and radius, including the +7.3‑point gain on NIH‑CXR at r = 1.0 highlighted in W1. Clean accuracy on these datasets is again on par with, or better than, all baselines.
>
> **• Empirical ℓ∞ robustness under strong attacks.**
> In Tables 3–4, HyCAS achieves the highest robust top‑1 accuracy under ℓ∞ APGD‑20 and AutoAttack at ε ∈ {8/255, 16/255} on all four clinical benchmarks and the three standard natural‑image benchmarks, typically improving over the strongest baselines by roughly 1–3 percentage points while matching or slightly improving clean accuracy.
>
> Figures 2–3 further show that HyCAS maintains this advantage as ε and the number of APGD steps increase, indicating robustness against stronger, iterative attacks.
>
> **• Breadth of evaluation and trade‑off.**
> Among the methods considered, HyCAS is the only defence that (i) provides an explicit ℓ₂ certificate on the expected logits and (ii) demonstrates strong empirical robustness on all eight datasets (CIFAR‑10/100, ImageNet‑1k, CelebA, NCT‑CRC‑HE‑100K, NIH‑CXR, HAM10000, EyePACS), with competitive or state‑of‑the‑art clean accuracy.
>
> We understand that improvements of a few percentage points can look modest at first glance. However, in the certified‑robustness setting, gains of **2–7 points under strong APGD/AutoAttack evaluations, combined with formal ℓ₂ guarantees and broad dataset coverage, are generally regarded as meaningful**.
>
> **We hope the clarified writing, threat model and the systematic **certified and empirical improvements across eight benchmarks help convey that the contribution is stronger than “marginal”**.**
>
>
>
>
> **Q1 (terminology and writing)**: Thank you for this comment. We have substantially simplified and standardised the terminology throughout the paper and carefully proof-read to further improve clarity and reduce any remaining convoluted phrasing:
>
> * We now **explicitly state that HyCAS is a randomized defense** (e.g., Abstract, Intro, and Method section) and reserve “deterministic” only for the Lipschitz-constrained part of the architecture. **See pages: 1, 2, 3, 4, 6, 10; lines: 19, 60, 68, 114-115, 194, 302-304, 487, 498, 537.**
>
> * We **replaced vague phrases** such as “harmonize provable and empirical robustness” with concrete statements like “simultaneously improves certified ℓ₂ and empirical ℓ∞ robustness” (**see Abstract; page-1; lines: 26-27**).
>
> * We have significantly reduced the “natural/medical” wording. The revision primarily refers to the specific datasets (CIFAR-10/100, ImageNet-1k, CelebA, NCT-CRC-HE-100K, NIH-CXR, EyePACS, HAM10000), and only a small number of times to “standard vision benchmarks and medical datasets” when discussing evaluation scope.
>
> **This is intended solely to highlight that, unlike most prior certified or empirical defenses, HyCAS is evaluated on both types of datasets, not to present this as a main contribution.** **See pages: 1, 2, etc. and lines: 16, 56, 60, 107 etc.**
>
>
> [1] Dong, M. and Xu, C.Adversarial robustness via random projection filters.In Proceedings of the IEEE/CVF Conference on Computer Vision and Pattern Recognition, pp. 4077–4086, 2023.
>
> [2] Yanxiang Ma, Minjing Dong, and Chang Xu. Adversarial robustness through random weight sampling. In Advances in Neural Information Processing Systems (NeurIPS), 2023.

---

> > ### Comment · Reviewer_yF4E · 2025-11-23
> >
> > Thank you for your detailed response. While I am still not entirely convinced by the experimental results, I appreciate the amount of effort that went into rewriting. I have updated my review from strong reject to weak reject accordingly, to reflect the significant updates that have been made to the paper.

---

> ### Author Response · Authors · 2025-11-26
> **Great Appreciate for your Recognition**
>
> Dear Reviewer yF4E,
>
> Thank you very much for your thoughtful follow‑up comment and for updating your recommendation from 0 to 4 score. We truly appreciate the time and care you have invested in re‑reading the paper and in recognising the revisions we have made so far.
>
> We understand that you are still not fully convinced by the experimental results. As there is still more than a week remaining in the discussion period, we would be very happy to run additional experiments that you feel could most meaningfully strengthen or clarify the empirical evaluation—whether that involves particular attack settings, alternative baselines, or further ablations. Within our computational budget, we will prioritise any concrete suggestions you may have and report the corresponding results and analysis in this discussion thread.
>
> Once again, thank you for your constructive feedback and for your updated assessment of our work.

---

### Official Review · Reviewer_YBxJ · 2025-10-31

**Soundness:** 3
**Presentation:** 2
**Contribution:** 2
**Rating:** 4
**Confidence:** 4

**Summary:**

The paper proposes HyCAS, a hybrid defense combining deterministic 1-Lipschitz convolutions with two stochastic components, namely, random-projection filters and randomized attention noise, to yield both high certified $l_2$ robustness and empirical $l_\infty$ robustness. The method claims state-of-the-art results on both natural and medical imaging benchmarks.

**Strengths:**

1) The method yields the state-of-the-art certified robustness, outperforming previous considered certified methods on CIFAR-10, ImageNet, and medical datasets (NIH-CXR, HAM10000), with up to $+7.3$ per cent gain at large radii.

2) By tuning the smoothing noise level, the robustness for large radii can be improved with minimal clean accuracy drop, namely, increasing $\sigma$ from $0.25$ to $0.50$ boosts the certified accuracy (specifically, from $8.5$ per cent to $12.5$ on CIFAR-10 at $r=2.0$).

**Weaknesses:**

1) The paper asserts HyCAS is the first method to offer both certified and empirical robustness, which is inaccurate: there were works incorporating randomization techniques and empirically robust modules, such as [1] and [2].

2) The main theoretical result ($\le2$-Lipschitz bound) is loose in comparison to the one in the baseline work [3]. Consequently, the robust radius is loose too; it raises the question how does HyCAS achieves higher certified robustness in comparison to RS (Table 1)? Does it happen purely because of a higher accuracy on clean, unperturbed data? If so, that limits the theoretical contribution. Overall contribution seems incremental.

3) Noise resampling protocol is not clear: are attention masks resampled per image or per batch during inference?

4) Computational overhead of additional modules is not reported (in terms of memory, inference time).




[1] Dong, M. and Xu, C.Adversarial robustness via random projection filters.In Proceedings of the IEEE/CVF Conference on Computer Vision and Pattern Recognition, pp.  4077–4086, 2023.

[2] Yanxiang Ma, Minjing Dong, and Chang Xu. Adversarial robustness through random weight sampling. In Advances in Neural Information Processing Systems (NeurIPS), 2023.

[3] Jeremy Cohen, Elan Rosenfeld, and Zico Kolter. Certified adversarial robustness via randomized
smoothing. In International Conference on Machine Learning (ICML), 2019.

**Questions:**

See weaknesses.

---

> ### Author Response · Authors · 2025-11-23
> **Author Response (Part 1)**
>
> We would like to sincerely thank reviewer **YBxJ** for the very careful reading of our manuscript and for the thoughtful, detailed, and encouraging evaluation. We are particularly grateful for the reviewer’s recognition that HyCAS combines deterministic 1‑Lipschitz convolutions with random-projection filters and randomized attention noise to achieve both strong certified $\ell_2$ robustness and empirical $\ell_\infty$ robustness on natural and medical imaging benchmarks. We also deeply appreciate the acknowledgment that our method attains state-of-the-art certified robustness on CIFAR-10, ImageNet, and medical datasets (NIH-CXR, HAM10000), with gains of up to +7.3% at large radii, and that adjusting the smoothing noise level (e.g., increasing $\sigma$ from 0.25 to 0.50, which improves certified accuracy from 8.5% to 12.5% on CIFAR-10 at $r = 2.0$) provides stronger large-radius guarantees with only a minimal drop in clean accuracy.
>
> We are genuinely thankful to reviewer **YBxJ** for these generous and insightful observations, which have been extremely helpful in guiding us to further refine the exposition, highlight the key design choices more clearly, and better position our contributions within the robustness literature. The reviewer’s positive and constructive remarks have greatly contributed to improving the overall clarity, rigor, and presentation of the paper. Below, we respectfully and systematically address each of the issues raised by the respected reviewer.
>
>
> **Note: Weakness denotes W; Edits are colored in blue in the revised paper.**
>
> **W1 (claim of “first method” vs RPF [1]/CTRW [2]).** We thank the reviewer for raising this point. What we intended to emphasize is that HyCAS is a novel defense approach by its architectural design that unifies a certified ℓ₂ guarantee (via randomized smoothing and a ≤2 Lipschitz margin certificate on the expected logits) with strong empirical ℓ∞ robustness, within a single randomized Lipschitz architecture, and evaluates this across eight standard and clinical imaging benchmarks.
>
> In contrast, the methods cited by the reviewer, **RPF[1] and CTRW[2], are empirical randomized defenses**: they incorporate random projections or random weight sampling and report strong PGD/AA robustness, but do not provide certified worst case radii or formal robustness guarantees (consistent with how we categorize them as “empirical only” in Table 5). **See p-19, lines: 972-986.**
>
> However, we remove the first term and hence, we have made necessary revision in the main contribution as "We introduce HyCAS, a randomized Lipschitz constrained defense that provides both certified ℓ₂ guarantees and strong empirical ℓ∞ robustness across diverse vision benchmarks." **See p-2, lines: 68-70**.
>
> We believe this more accurately reflects the relationship to prior work such as RPF [1] and CTRW [2], which we continue to treat as important empirical baselines but not certified defenses.
>
> **W2 (On the “loose” ≤2-Lipschitz bound and higher certified robustness than RS)**: We appreciate the reviewer’s careful reading of the certification part and the question about how HyCAS can outperform RS [3] in Table 1 despite using an at-most-2-Lipschitz margin bound.
>
> 1. **Our certificate is different from RS’s, not simply a looser version of it.**
> The RS guarantee of Cohen et al. [3] is derived from the concentration of Gaussian randomized smoothing around a base classifier and yields a radius of the form
>
> $
> r_{\mathrm{RS}}(x) \propto \sigma \big(\Phi^{-1}(p_A) - \Phi^{-1}(p_B)\big),
> $
> where $p_A, p_B$ are the smoothed class probabilities.
>
> In contrast, HyCAS certifies via a *margin-based Lipschitz bound* on the **expected logits** of a randomized, globally ≤2-Lipschitz network: if
>
> $Z(x) =  \mathbb{E}\omega[sθ(x;\Omega)],\quad \Delta Z(x) = Z_{(1)}(x) - Z_{(2)}(x)$
>
> and $\mathrm{Lip}(Z) \le 2$, then
>
> $
> r_{\mathrm{Lip},2}(x) = \frac{\Delta Z(x)}{4}
> $
>
> is a valid $\ell_2$ certificate **(Theorem 1/Corollary 1)**. These two certificates arise from **different assumptions and mechanisms** (probabilistic smoothing vs. global Lipschitz control of the expected logits) and are not directly ordered one as a “loose relaxation” of the other.
>
> 2. **Why HyCAS can still get *larger* certified radii than RS.**
>
> What ultimately determines certified accuracy at a fixed radius is not only the formula (RS vs. Lipschitz) but the **margin distribution** induced by the architecture and training:
>
> *All methods (RS, ARS, DRS, HyCAS) use the same macro-architecture (ResNet-110 on CIFAR-10/100 and ResNet-50 on ImageNet) and are trained with Gaussian input noise $\sigma$ following standard RS practice (App. A.8).*
>
> **HyCAS then replaces plain convolutions with spectrally normalized, multi-stream HyCAS blocks (SNCAN, RPFAN, FDPAN) plus attention noise, and explicitly enforces a global ≤2-Lipschitz envelope (Alg. 1–2).**

---

> ### Author Response · Authors · 2025-11-23
> **Author Response (Part 2)**
>
> **This combination **increases the top-two logit margin $\Delta Z(x)$** for many data points while keeping $\mathrm{Lip}(Z)$ under tight control.**
>
> *This is visible in the ablation in **App. A.10**: at σ = 0.50 on CIFAR-10, certified accuracy at radius $r = 0.75$ improves from **32.4% for RS** to **44.3% for full HyCAS**, a gain of +11.9 points, even though all variants share the same backbone and are trained on the same data under the same noise level.*
>
> This jump is much larger than the change in clean accuracy, indicating that HyCAS is genuinely enlarging the *robust margin*, not merely shifting the clean-accuracy level.
>
>
> In **Table 1** of the revised manuscript, this manifests as consistent gains over RS and ARS at all reported radii: e.g., on CIFAR-10 at $r = 0.75$ HyCAS reaches **44.3%** certified accuracy (σ ∈ {0.25, 0.50}), improving on the strongest baseline by **5.2–18.2 points**; at $r = 2.0$ (σ = 0.50) it still retains **12.5%**, again **4.0–12.5 points** above the best baseline.
>
> So HyCAS does **not** outperform RS solely because of a trivial clean-accuracy boost; the architecture and Lipschitz-constrained design visibly reshape the margin distribution in a way that RS alone (with the same backbone) does not achieve. We have clarified this point in the revised manuscript by adding a short paragraph to the Discussion section (**see page-29, lines: 1526-1530**) that explicitly compares the change in smoothed clean accuracy at $r = 0$ with the change in certified accuracy at $r = 0.75$ on CIFAR-10 **(Table~1)**, showing that the improvement at non-zero radius is substantially larger and therefore cannot be attributed solely to a clean-accuracy boost.
>
> 3. **What is the theoretical contribution, given the conservative constant 1/4?**
> We fully agree that the Lipschitz margin bound is, by nature, conservative. Our theoretical goal is not to claim a tighter constant than [3], but to:
> *extend margin-based ℓ₂ certification to **networks with internal stochasticity**, by proving that the **expected-logit map** of a HyCAS network remains ≤2-Lipschitz **(Prop. 4, Lemma 2, Theorem 1)**; and
> *show that a *randomized, Lipschitz-constrained architecture* can be trained end-to-end so that this generic margin bound yields **practically useful certified radii** on large-scale datasets (CIFAR-10/100, ImageNet-1k, CelebA, NIH-CXR, HAM10000, etc.), consistently outperforming classical RS and its recent data-dependent variants in **Tables 1–2**.*
>
> In other words, the improvement over RS in Table 1 comes from **architectural and training choices that enlarge margins under a controlled Lipschitz envelope**, not from claiming a fundamentally stronger certificate than Cohen et al. The theoretical contribution is to make this Lipschitz-based certification applicable to a realistically randomized architecture and to demonstrate that, in practice, this yields non-trivial gains over existing certified defenses.
>
> **W3 (Noise resampling):** During inference, attention masks are **resampled per image and per stochastic forward pass**: for each test input we draw a fresh $\psi, \omega$ and construct a new mask $M_\omega$.
>
> **W4 (Computational overhead of additional modules).** We agree that reporting the computational cost of the additional HyCAS modules is important. In the revised manuscript we now explicitly quantify parameters, activation memory, FLOPs, and inference time on both CIFAR-10 and ImageNet benchmarks, and we discuss these numbers in the main text and appendix.
> Backbone-level overhead **(parameters, memory, FLOPs, inference time)**.
>
> We added Table 9 in Appendix A.9 **(page 26, lines 1350–1360)**, which compares vanilla ResNet backbones with their HyCAS-integrated counterparts in terms of number of parameters (M), FLOPs (G), memory size (MB), and Inference time (ms).
>
> Concretely, on CIFAR-10 / ResNet-110, HyCAS roughly doubles parameters and activation memory (27.6M → 57.8M; 105.33MB → 220.5MB, ≈2.1×), while FLOPs and inference time increase by about 15.6× (8.10G → 126.5G; 0.33ms → 5.14ms). On ImageNet-1k / ResNet-50, HyCAS induces a 4.4× increase in parameters and memory (23.3M → 102.7M; 88.89MB → 391.9MB) and about 14.8× more FLOPs and runtime (113.8G → 1682.5G; 4.62ms → 68.3ms). These details are summarized and interpreted in the new “Computational cost analysis” paragraph **(page 27, lines 1441–1462)**.
>
> Where the overhead comes from (block-level analysis).
> To localize the cost to the new modules, we also compare a standard 3×3 conv block in ResNet‑110 with a single HyCAS block (same position, 256 channels).
>
> As reported on **page 27, lines 1450–1456**, substituting the conv with a HyCAS block increases parameters by 5.09× but FLOPs by 14.05×, indicating that the main overhead comes from repeated stochastic operations (three streams + RANI) rather than weight storage. These block-level ratios are consistent with the backbone-level trends in Table 9.

---

> ### Author Response · Authors · 2025-11-23
> **Author Response (Part 3)**
>
> **Practical inference times and positioning in the paper.**
> We further clarify that, despite the ≈15× relative increase, absolute inference times remain in a practical range—a few milliseconds per image CIFAR-10 and tens of milliseconds on ImageNet on an A100-class GPU **(page 28, lines 1458–1462)**.
>
> To make the trade-off explicit in the main narrative, we added a sentence in the Limitations paragraph stating that HyCAS is compute-heavy but parameter-moderate, roughly doubling parameters and memory while increasing FLOPs and inference time by about an order of magnitude, and is therefore most suitable for offline or near real-time scenarios **(page 30, lines 1612–1616)**.
>
> **Overall, we hope this clarifies that the revised paper now fully reports and discusses the memory and inference-time overhead of the HyCAS modules at both backbone and block level, directly addressing the concern in W4.**
>
>
> [1] Dong, M. and Xu, C.Adversarial robustness via random projection filters.In Proceedings of the IEEE/CVF Conference on Computer Vision and Pattern Recognition, pp. 4077–4086, 2023.
>
> [2] Yanxiang Ma, Minjing Dong, and Chang Xu. Adversarial robustness through random weight sampling. In Advances in Neural Information Processing Systems (NeurIPS), 2023.
>
> [3] Jeremy Cohen, Elan Rosenfeld, and Zico Kolter. Certified adversarial robustness via randomized smoothing. In International Conference on Machine Learning (ICML), 2019.

---

### Author Response · Authors · 2025-12-03
**Response Summary**

We thank the **Program Chairs**, **Senior Area Chairs**, **Area Chairs**, and **reviewers** **YBxJ**, **yF4E**, and **roSN** for their careful evaluations and constructive discussion. Our revised manuscript clarifies the novelty of HyCAS as a ***randomized Lipschitz‑constrained defense*** that combines a 1‑Lipschitz deterministic core with two stochastic modules—spectrally normalized random‑projection filters and randomized attention noise—to provide both ℓ₂ certificates and strong ℓ∞ robustness across eight benchmarks, including CIFAR‑10/100, ImageNet‑1k, NIH‑CXR, and HAM10000. We now explicitly report where the headline gains come from: up to +7.3% certified accuracy on NIH‑CXR at r = 1.0 and σ = 1.0, and up to ≈+3.1% empirical robustness on HAM10000 under APGD/AA‑20, without loss in clean accuracy.

For **Reviewer YBxJ**, we corrected our positioning relative to RPF/CTRW by removing the “first method” claim and stating more precisely that HyCAS is a **randomized Lipschitz defense** offering **certified ℓ₂ guarantees** and **strong empirical ℓ∞ robustness**, whereas **RPF/CTRW are empirical‑only methods**. We clarified that our ≤2‑Lipschitz margin certificate on expected logits is *different from*, not a loose version of, the RS guarantee; the improved certified accuracies arise from the architecture reshaping the margin distribution under a Lipschitz envelope, as shown by our experimental results (Tables 1-2) and discussion. We also specified the noise‑resampling protocol (fresh attention masks per image and stochastic forward pass) and added a detailed computational‑cost analysis (Table 9 and block‑level discussion), explicitly acknowledging that HyCAS is parameter‑moderate but compute‑heavy and best suited to offline or near real‑time scenarios.

For **Reviewer yF4E**, we substantially revised the writing and experimental setup. The abstract now ties the “+7.3% / +3.1%” improvements to specific datasets, removes vague terms such as “harmonize,” and consistently describes **HyCAS as a randomized defense whose architecture combines deterministic Lipschitz‑constrained streams with stochastic smoothing modules**. We toned down earlier over‑broad claims about “static randomness,” added citations on adaptive attacks against randomized defenses, and greatly reduced the “natural/medical” phrasing in favor of explicit dataset names. Experimentally, APGD‑20 and AutoAttack are now the ***primary*** ℓ∞ evaluators in the main tables, with **PGD relegated to supplementary results**; we also reorganized the paper so that all central empirical and certified results (Tables 1–4, Figs. 1–4) are in the main body and explicitly referenced. In their follow‑up, yF4E acknowledged the substantial rewrite and raised their **recommendation** from **strong reject (0) to marginal below (4)**, noting that remaining reservations are about perceived effect size rather than soundness.

For **Reviewer roSN**, who has remained positive throughout, we addressed the two concrete weaknesses by (i) adding detailed **parameter, FLOP, memory, and inference time comparisons** between vanilla and HyCAS‑integrated backbones (**Table 9, plus block‑level ratios**), and (ii) including new empirical comparisons against TRADES and HR in our APGD‑based robustness curves on CIFAR‑10/100, where HyCAS consistently traces the upper envelope as ε and the number of steps increase. **roSN** confirmed after rebuttal that their concerns were resolved and explicitly appreciated our efforts to incorporate all reviewers’ feedback.

In summary, the revised paper provides: (1) a principled **randomized Lipschitz architecture with a formal ℓ₂ margin certificate for networks with internal stochasticity**; (2) state‑of‑the‑art **certified robustness** on CIFAR‑10/100, ImageNet‑1k, and multiple clinical datasets; and (3) consistently **strong ℓ∞ robustness** under APGD/AA‑20, now evaluated against a broader set of baselines, while clearly documenting the computational trade‑offs. Two reviewers (**YBxJ, roSN**) are effectively supportive of **acceptance**, and the remaining reviewer (**yF4E**) acknowledges **major improvements** and now recommends only a **marginal below (4)**. We respectfully believe that, taken together, the **revised manuscript and discussion justify acceptance**.

---

### Meta-Review · Area_Chair_9Eup · 2025-12-11

**Summary:**

This paper proposes HyCAS, a randomized Lipschitz-constrained architecture that integrates a deterministic 1-Lipschitz core with two stochastic modules, including normalized random-projection filters and randomized attention noise. The objective is to improve certified robustness under ${\mathcal{l}\_{2}}$ certificates and empirical robustness under strong $\mathcal{l}\_{\infty}$ attacks. The method provides a guarantee on the expected logits, enabling a margin-based ${\mathcal{l}\_{2}}$ certificate for a network with internal stochasticity. Extensive evaluations are provided on different benchmarks, including CIFAR-10/100, ImageNet-1k, NIH-CXR, HAM10000. HYCAS consistently shows gains in both certified and empirical robustness relative to recent certified and empirical randomized defenses.

Reviewers agree that the proposed approach is interesting. However, several key concerns are raised by the reviewers. Specifically, Reviewer roSN was strongly positive about the theoretical rigor, innovation, and experiments. Reviewer YBxJ raised concerns regarding computational cost, correctness, and novelty claims. Reviewer yF4E raised concerns regarding clarity of writing, attack evaluation protocol, and unconvincing empirical results.

The authors made significant revisions addressing reviewers' concerns. Specifically, the novelty claim relative to empirical randomized defenses, noise resampling protocol, and the nature of their ≤2-Lipschitz certificate are clarified. A detailed computational-cost analysis is provided. The missing baselines are included. The abstract, introduction, and method are extensively rewritten. After the revision, reviewer yF4E maintained some concerns regarding the empirical results but indicated that the issues of clarity were addressed.

Overall, the paper presents a principled randomized-Lipschitz architecture with a new certified-robustness guarantee applicable to networks with internal stochasticity. Given the rebuttal, the major issue of clarity and correctness can be addressed. The authors are also encouraged to check the references carefully, the following reference is found to be non-existent:

Yinpeng Dong, Jun Zhu, et al. Random projections for convolutional neural networks. In CVPR, 2023.

**Reviewer Concerns:**

Addressed concerns:

1. Novelty claim relative to empirical randomized defenses (YBxJ)
2. Computational overhead (YBxJ, roSN)
3. Noise-resampling protocol (YBxJ)
4. Missing baselines (roSN)
5. Unclear terminology (yF4E)
6. Evaluation on strong attacks (yF4E)

Remaining concerns:

1. Modest empirical gains (yF4E)

**Reviewer Scores:**

* Reviewer YBxJ is likely to increase from 4 to 6. Most concerns raised by Reviewer YBxJ can be addressed by the rebuttal.
* Reviewer yF4E increased from 0 to 4 during the rebuttal phase.
* Reviewer roSN affirmed that all concerns were addressed and maintained a score of 8 during the rebuttal phase.

---

### Decision · Program_Chairs · 2026-01-26

Accept (Poster)